# Arachidonic acid-regulated calcium signaling in T cells from patients with rheumatoid arthritis promotes synovial inflammation

Zhongde Ye[1,2], Yi Shen[2], Ke Jin[2], Jingtao Qiu [2], Bin Hu [1,2], Rohit R. Jadhav [1,2], Khushboo Sheth [1], Cornelia M. Weyand [1,2] & Jörg J. Goronzy [1,2✉]

Rheumatoid arthritis (RA) and psoriatic arthritis (PsA) are two distinct autoimmune diseases that manifest with chronic synovial inflammation. Here, we show that CD4[+] T cells from patients with RA and PsA have increased expression of the pore-forming calcium channel component ORAI3, thereby increasing the activity of the arachidonic acid-regulated calcium-selective (ARC) channel and making T cells sensitive to arachidonic acid. A similar increase does not occur in T cells from patients with systemic lupus erythematosus. Increased *ORAI3* transcription in RA and PsA T cells is caused by reduced IKAROS expression, a transcriptional repressor of the *ORAI3* promoter. Stimulation of the ARC channel with arachidonic acid induces not only a calcium influx, but also the phosphorylation of components of the T cell receptor signaling cascade. In a human synovium chimeric mouse model, silencing ORAI3 expression in adoptively transferred T cells from patients with RA attenuates tissue inflammation, while adoptive transfer of T cells from healthy individuals with reduced expression of IKAROS induces synovitis. We propose that increased ARC activity due to reduced IKAROS expression makes T cells more responsive and contributes to chronic inflammation in RA and PsA.

[1] Department of Medicine, Palo Alto Veterans Administration Healthcare System, Palo Alto, CA, USA. [2] Department of Medicine, Stanford University, Stanford, CA, USA. ✉email: jgoronzy@stanford.edu

Rheumatoid arthritis (RA) and psoriatic arthritis (PsA) are systemic autoimmune diseases presenting with chronic synovial inflammation that can result in the destruction of the joint architecture. Clinically, these diseases are clearly distinct. PsA occurs in the setting of psoriasis, an autoimmune skin disease characterized by the excessive and rapid growth of the epidermal layer as a result of skin inflammation. Joint manifestations occur in a subset of patients and are highly variable, frequently exhibiting the pattern of a spondyloarthropathy. B cell autoimmunity is not a hallmark of PsA. By contrast, a characteristic feature of RA is autoantibodies (auto-Abs) specific to Fc fragment of IgG, the so-called rheumatoid factors, and Abs to citrullinated self-antigens.

A common denominator of both diseases is that an abnormal CD4[+] T cell response has been implicated in the pathogenesis of the disease. Many genetic susceptibility factors of RA are related to T cell function. *HLA-DRB1* alleles dominate the genetic risk accounting for one-third of heritability[1]. Disease susceptibility has been mapped to positions 67 to 74 of the α-helix encoded by different *HLA-DRB1* alleles, frequently referred to as shared epitope[2]. Homozygosity for this relevant HLA-DRB1 structure is associated with increased risk and disease severity, which may indicate a critical role for T cell receptor (TCR) signaling strength in disease pathogenesis[3]. Non-HLA variants identified in genome-wide association studies explain approximately another 15% of heritability[4] and include many molecules that are preferentially expressed in T cells. Likewise, disease susceptibility genes of PsA are frequently involved in T cell biology; in particular, gene associations have been found related to interleukin-23 signaling that may explain the preferential activation of TH17 cells in the disease[5]. While many of the susceptibility genes are overlapping between cutaneous psoriasis and PsA, some of them are unique for PsA including several HLA variants[6].

The best example of an autoimmunity susceptibility gene involved in T cell activation is *PTPN22*[7]. The *PTPN22* 1858C > T polymorphism is the second-highest risk gene for RA. It is also associated with PsA, but not with psoriasis. Since it also confers risk for a number of other autoimmune diseases, it appears to be involved in critical tolerance mechanisms rather than disease-specific pathways. *PTPN22* encodes a protein tyrosine phosphatase. In addition to promoting pattern-recognition receptor-induced type I interferon production by myeloid cells, it regulates TCR signaling by forming a complex with the C-terminal SRC kinase, an inhibitor of TCR signaling[7]. The disease-associated polymorphisms impair this interaction; with currently no consensus on the functional consequences.

Understanding how self-tolerance in T cells fails continues to be one of the scientific challenges in autoimmune diseases[8,9]. Peripheral tolerance refers to the mechanisms of preventing an adaptive immune response, although self-reactive T cells are present in the periphery of all individuals and constitute a large fraction of the repertoire[10,11]. In addition to regulatory T cells[12–14], several immune checkpoints control T cell responses, including the expression of costimulatory and coinhibitory molecules; aberrant costimulation or lack of coinhibition has been shown to cause autoimmunity in model systems[15]. Abnormalities in T cell differentiation have been identified in RA and PsA, but the defect facilitating the activation of autoreactive T cells remains unknown.

RA patients share with PsA patients the increased transcription of the *ORAI3* gene. ORAI3 is a member of a family of ORAI molecules that form selective homomeric and heteromeric calcium ion channels. The regulation of channel activity depends on its chain composition. Homomeric ORAI1 forms the store-operated $Ca^{2+}$release-activated $Ca^{2+}$(CRAC) channel known to be important in T cell activation when TCR stimulation induces inositol (1,4,5) trisphosphate-mediated release of calcium ions from the endoplasmic reticulum (ER) leading to the recruitment of STIM1 (stromal interaction molecule 1) to ORAI1 and the influx of calcium into the cytoplasm. ORAI3 lacks three critical C-terminal residues that determine the interaction of ORAI1 with STIM1. A heteromeric channel including ORAI3 has a reduced calcium current, while still being store-operated through ORAI1. In contrast, ORAI3 is receptive to pharmacological modulation, its binding to arachidonic acid (AA) activates the heteromeric channel independent of intracellular stores[16–21]. The ORAI1/3 channel is therefore referred to as AA-regulated calcium-selective (ARC) channel, but can also be activated by leukotriene C4[22]. ORAI3 has gained increased interest because it is overexpressed in cancer, in particular estrogen receptor-expressing breast cancer cells, where it contributes to cancerous properties by inducing calcium influx, phosphorylation of ERK, and expression of c-myc[21,23–29].

Here, we show that RA T cells with increased ORAI3 expression respond to AA stimulation with increased $Ca^{2+}$ influx. This signal is sufficient to induce phosphorylation of components of the TCR signaling cascade and the expression of the TCR stimulation-specific response gene *NUR77*. This finding is reproduced in T cells by forced overexpression of ORAI3. Increased *ORAI3* transcription is due to a decline in IKAROS expression that functions as a repressor of the *ORAI3* promoter. We propose that the increased ARC responsiveness to endogenously produced AA is a shared pathogenetic pathway in the synovial inflammation of RA and PsA by making CD4[+] T cells more responsive to autoantigenic stimulation.

## Results

**Constitutive activation of TCR signaling in RA T cells**. To determine whether naive and therefore antigen-inexperienced T cells in RA patients show evidence of increased constitutive activation compared to those from healthy control (HC), we quantified Y128 phosphorylation of SLP76, a proximal scaffolding molecule phosphorylated by the ZAP-70 kinase immediately downstream of TCR stimulation, in freshly separated peripheral blood mononuclear cells (PBMCs). p-SLP76 was significantly increased in naive RA CD3[+]CD4[+]CD45RA[+]CD62L[+] T cells ($p = 0.01$, Fig. 1a, unpaired two-tailed Student's $t$ test). Consistent with SLP76 phosphorylation, we also found increased phosphorylation of ERK ($p = 0.01$, Fig. 1b, unpaired two-tailed Student's $t$ test). In these phosphoflow assays, PBMCs from HCs and RA patients were always analyzed in parallel to avoid experimental batch effects inherent to phosphoflow studies. In further support of constitutive activation, Fluo-8 staining indicated increased cytoplasmic calcium levels in RA naive CD4[+] T cells (Supplementary Fig. 1b), which was subsequently confirmed by ratiometric fluorescence studies with Fura Red ($p = 0.01$, Fig. 1c, unpaired two-tailed Student's $t$ test). Accordingly, constitutive mean fluorescence intensity (MFI) of CD69, an early and easily inducible activation marker was about twofold higher in RA CD3[+]CD4[+]CD45RA[+]CD62L[+] T cells ($p < 0.0001$; Fig. 1d, unpaired two-tailed Student's $t$ test). Moreover, we also found increased expression of *NUR77*, an early response gene that is considered to be specific for TCR stimulation and allows to distinguish antigen-induced from bystander activation ($p = 0.0006$, Fig. 1e, unpaired two-tailed Student's $t$ test)[30]. These activation markers could not be induced in T cells from healthy individuals by culture with RA serum (Supplementary Fig. 2). Collectively, these data suggest that naive CD4[+] T cells from RA patients display evidence of activation downstream of TCR triggering.

**Increased expression of ORAI3 in naive RA CD4[+] T cells**. To explore whether naive CD4[+] T cells from RA patients have a defect in calcium homeostasis causing the increased cytoplasmic

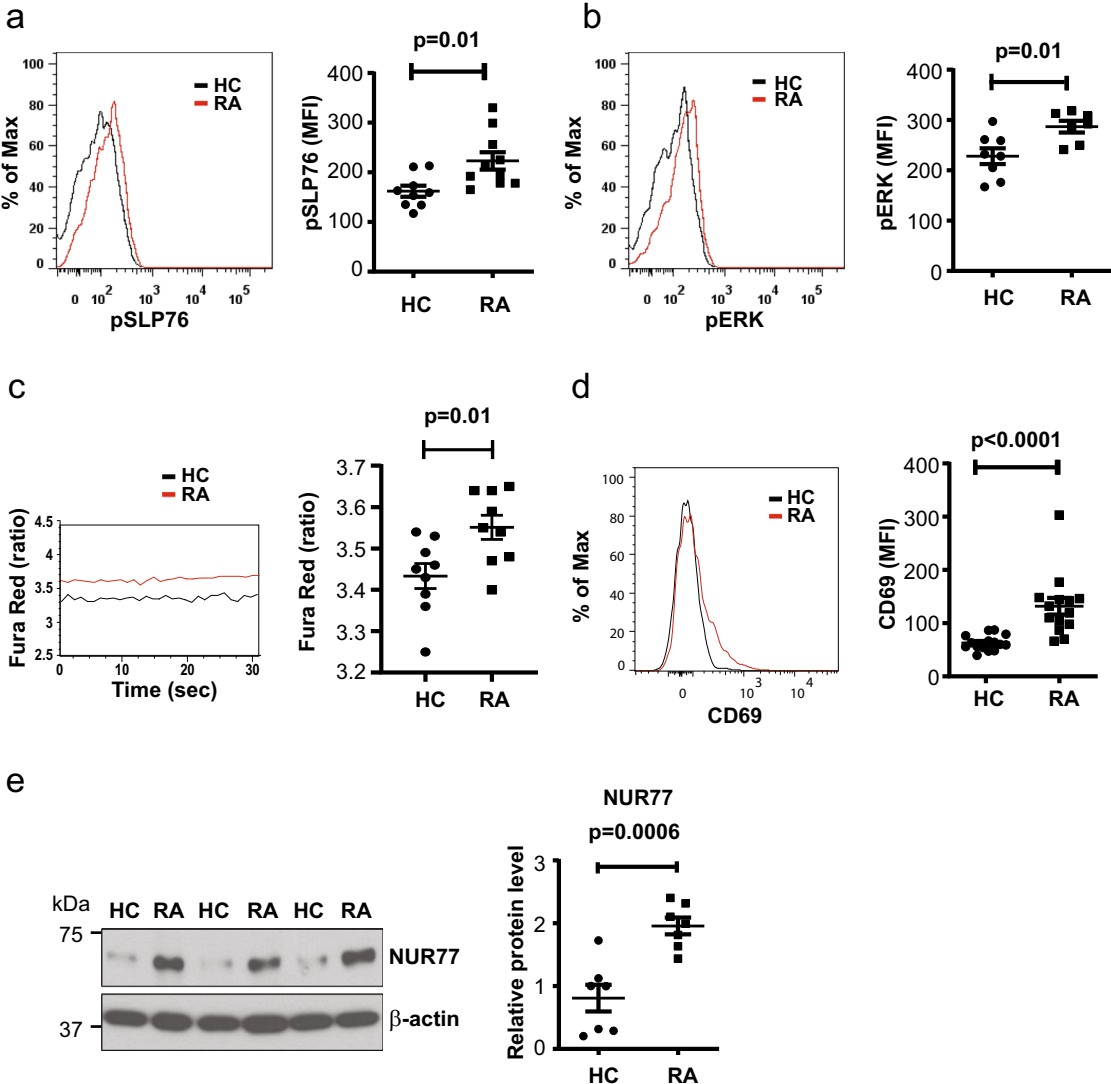

**Fig. 1 Constitutively activated TCR signaling in peripheral naive RA CD4+ T cells.** Peripheral blood mononuclear cells (PBMCs) were analyzed by flow cytometry directly ex vivo without prior in vitro stimulation. Samples of rheumatoid arthritis (RA) patients and healthy controls (HCs) were always run in parallel. **a** Representative histograms and mean fluorescence intensity (MFI) of phosphorylated SLP76 (Y128) in gated CD3+CD4+CD45RA+CD62L+ naive CD4+ T cells from HC ($n = 9$) and RA ($n = 10$) patients are shown. Gating strategies for naive CD4+ T cells is shown in Supplementary Fig. 1a. **b** Representative histograms and MFI of phosphorylated ERK (Thr202/Tyr204) in CD3+CD4+CD45RA+CD62L+ naive CD4+ T cells from HC ($n = 8$) and RA ($n = 7$) patients. **c** Representative tracing of Fura Red ratios (left) and shifts in Fura Red fluorescence (at 406 and 532 nm) in naive CD4+ T cells from HC ($n = 9$) and RA ($n = 9$) patients (right). **d** Representative histograms of CD69 expression in CD3+CD4+CD45RA+CD62L+ naive T cells (left) and data from HC ($n = 15$) and RA ($n = 14$) patients (right). **e** NUR77 expression in isolated naive CD4+ T cells shown as representative immunoblots (left) and plots of relative densities from HC ($n = 7$) and RA ($n = 7$) normalized to β-actin (right). Uncropped Western blots in Supplementary Fig. 9. All data are presented as dot plots with mean ± SEM. All statistical analyses were performed with unpaired two-tailed Student's t test. Source data are provided as a Source Data file.

concentrations shown in Fig. 1c, we profiled transcription of ten Ca²⁺ channel and sensor genes. In an initial screening cohort of six RA patients and six HCs, we found an increased transcription of *ORAI3*, while there was no difference in components and regulators of the classical CRAC channel (Fig. 2a). Over-expression was confirmed in a subsequent cohort of 22 HC and 21 RA patients ($p < 0.0001$, Fig. 2b, unpaired two-tailed Student's t test). No difference in ORAI3 expression between female (1.96 ± 0.22) and male RA patient (2.28 ± 0.28) and no correlation to age was seen (Supplementary Fig. 3). *ORAI3* transcription was independent of disease activity (Fig. 2c): transcript levels were already higher in RA patients with clinical disease activity index (CDAI) of <10 and did not significantly further increase in patients with CDAI > 20. Moreover, no influence of treatment

with methotrexate or tumor necrosis factor (TNF) inhibitors was seen (Fig. 2d, e). Increased expression of ORAI3 in RA naive CD4+ T cells was confirmed at the protein level by Western blotting (Fig. 2f). To determine whether the increase in *ORAI3* transcripts is disease-specific, we profiled naive CD4+ T cells from patients with gout, systemic lupus erythematosus (SLE), and PsA for the expression of ORAI3. *ORAI3* transcription was also increased in PsA, albeit not to the same extent as in RA, while the number of transcripts in naive CD4+ T cells from patients with SLE or gout were not different from those from HCs (Fig. 2g).

**Functional consequences of increased ORAI3 expression.** ORAI3, together with ORAI1, forms the ARC channel that

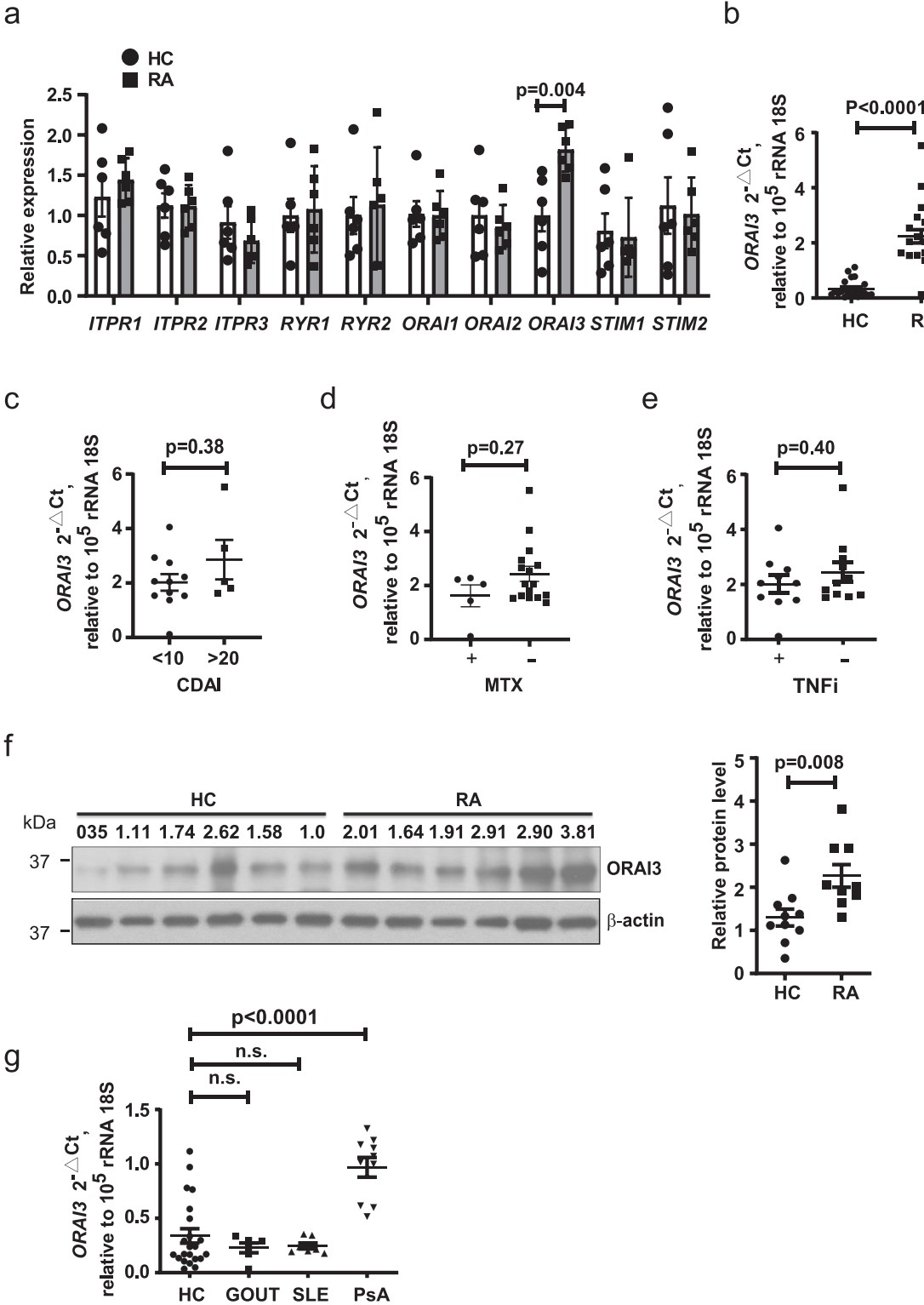

mediates agonist-activated, store-independent calcium entry[31], in contrast to the store-operated CRAC channel that mediates the $Ca^{2+}$ influx after TCR triggering and that is an exclusive ORAI1 polymer[32–34]. To determine whether the ARC channel is functional in naive $CD4^+$ T cells, we monitored cytoplasmic calcium concentrations after AA stimulation by ratiometric Fura Red fluorescence. We found a rapid influx of $Ca^{2+}$ ions upon stimulation (Fig. 3a). After the peak response, $Ca^{2+}$ concentrations declined again, but remained slightly elevated. This peak response

was smaller than the CRAC channel-mediated calcium influx after TCR stimulation, but it was sufficient to induce Thr286 phosphorylation of CAMKII (Fig. 3b). Unexpectedly, AA also induced phosphorylation of the CD3ζ chain (Fig. 3c) and induction of NUR77 expression in the absence of any additional TCR triggering (Fig. 3d). Moreover, stimulation with increasing concentrations of AA followed by culture for 24 h in the absence of exogenous AA or additional TCR stimulation resulted in a dose-dependent expression of CD69 (Fig. 3e).

**Fig. 2 Increased expression of the Ca$^{2+}$ channel component ORAI3 in RA CD4$^+$ T cells. a** Transcriptional profiling of Ca$^{2+}$ channel and Ca$^{2+}$ sensor gene transcripts in CD4$^+$ naive T cells from HC ($n = 6$ white bars and circles) and RA ($n = 6$ gray bars and squares) patients. Gene expression was normalized to β-actin. Data are presented as mean ± SEM. **b** *ORAI3* transcripts quantified by qPCR and normalized to 18S rRNA in CD4$^+$ naive T cells from HC ($n = 22$) and RA ($n = 21$). Data are shown as dot plots with mean ± SEM. **c** *ORAI3* transcript levels in RA patients with CDAI activity scores <10 ($n = 9$) or >20 ($n = 5$). **d**, **e** *ORAI3* transcripts in CD4$^+$ naive T cells from RA patients dichotomized for whether they were treated ($n = 5$) or not treated with methotrexate (MTX) ($n = 16$) (**d**) or whether they were ($n = 10$) or were not ($n = 11$) on TNF inhibitors (TNFi, etanercept or adalimumab) (**e**). **f** ORAI3 protein levels are shown as a representative immunoblot (left) and as dot plots of densities of ORAI3 relative to β-actin of purified naive CD4$^+$ T cells from HC ($n = 10$) and RA ($n = 9$) patients. Uncropped Western blots in Supplementary Fig. 9. **g** *ORAI3* transcripts quantified by qPCR and normalized to 18S rRNA in CD4$^+$ naive T cells from HC ($n = 22$) and gout ($n = 6$), systemic lupus erythematosus (SLE) ($n = 7$), and PsA ($n = 10$) patients (HC vs. gout $p = 0.80$; HC vs. SLE $p = 0.85$). Statistical analyses were performed with one-way ANOVA followed by Tukey's multiple comparison test (Fig. 2g) or with unpaired two-tailed Student's $t$ test (**a–f**). n.s. not significant. Source data are provided as a Source Data file.

To confirm that the AA effect is mediated by the ARC channel, we transfected naive CD4$^+$ T cells with small interfering RNA (siRNA) smart pool for ORAI3. Reduced ORAI3 expression attenuated the AA-induced calcium influx (Fig. 4a), but did not affect thapsigargin-induced calcium influx (Supplementary Fig. 5). Moreover, lentivirus transduction of CD4$^+$ naive T cells with two different short hairpin RNAs (shRNAs) targeting different sequences of ORAI3 inhibited phosphorylation of ERK after AA stimulation (Fig. 4b). Likewise, ORAI3 silencing in Jurkat cells reduced the constitutive expression of CD69 (Fig. 4c) and NUR77 (Fig. 4d). Conversely, forced overexpression of ORAI3 in Jurkat cells increased AA-induced calcium influx and ERK phosphorylation (Fig. 4e, f) as well as constitutive expression of CD69 (Fig. 4g). Taken together, Ca$^{2+}$ signaling controlled by AA is dependent on the concentration of ORAI3, presumably by increased ARC channel activity. Stimulation of the ARC channel by AA not only resulted in the expected calcium fluxes but was also sufficient to induce phosphorylation of signaling components in the TCR signaling pathway and the expression of early response genes.

**Increased AA-dependent Ca$^{2+}$ signaling in RA T cells**. To determine whether the increased ORAI3 expression in RA has functional consequences, we compared the response of naive CD4$^+$ T cells from HC and RA patients to AA stimulation. Peak responses of calcium influxes were significantly higher in RA T cells compared to HC ($p = 0.002$, Fig. 5a, unpaired two-tailed Student's $t$ test). In addition, AA-induced ERK phosphorylation was increased (Fig. 5b). Consistent with the results described in Fig. 1d, RA CD4$^+$ naive T cells had higher baseline levels of CD69, which were further increased by AA stimulation (Fig. 5c). Naive CD4$^+$ T cells from PsA patients had similarly increased responses to AA stimulation as RA T cells (Fig. 5c). To determine whether endogenous AA production accounted at least, in part, for the increased baseline CD69 expression in patients, we treated naive CD4$^+$ T cells with increasing concentrations of the phospholipase A2 (PLA2) inhibitor AACOCF3[35] and found a dose-dependent effect (Fig. 5d). Treatment with 6 μM AACOCF3 reduced the constitutive expression of CD69 and NUR77 in naive RA and PsA CD4$^+$ T cells (Fig. 5e, f). Serum concentrations of AA were not elevated in our cohorts of RA or PsA patients (Supplementary Fig 4), suggesting that the increased expression of ORAI3 and not increased production of AA accounted for the increased ARC channel activity.

**Silencing of ORAI3 reduces synovial inflammation**. Previous studies in a human synovium-NSG (NOD.Cg-Prkdcsci-dIl2rgtm1Wjl/SzJ) mice chimera system have shown that naive CD4$^+$ T cells from RA patients drive synovial inflammation more vigorously than those from HC[36–38]. To determine whether increased ORAI3 expression in RA T cells enables the inflammatory response, NSG mice engrafted with human synovial tissue were reconstituted with CD45RO$^-$ PBMC from RA patients transfected with control or ORAI3 siRNA. ORAI3 silencing markedly reduced the overall density of the synovial T cell infiltrate ($p < 0.0001$, unpaired two-tailed Student's $t$ test) and the frequencies of T cells producing interferon-γ (IFN-γ) in situ ($p < 0.0001$, Fig. 6a, unpaired two-tailed Student's $t$ test). In addition to the histological studies, synovial inflammation was quantified by gene expression analysis of TCR genes, lineage-specific transcription factors and key inflammatory mediators. Silencing ORAI3 expression in adoptively transferred T cells reduced TRB transcripts consistent with the histological studies. TH1 and TH17 T cells were reduced as can be concluded from the decreased expression of TBX21 and RORγt and the reduced production of RANKL (*TNFSF11*), IFN-γ, IL-17A, and TNF (Fig. 6b). In contrast, FOXP3 transcripts, indicating the presence of Treg, did not change, Production of IL-1β and IL-6 were attenuated, likely downstream of the reduced T cell activity.

**Transcriptional repression of *ORAI3* by IKAROS**. To examine the transcriptional regulation of *ORAI3*, we first identified sites of chromatin accessibility using ATAC-seq (assay for transposase-accessible chromatin using sequencing). As shown in Fig. 7a, the accessibility profiles at the *ORAI3* gene in naive CD4$^+$ T cells from RA, PsA, and HC were not different, with the transcription start site (TSS) and additional downstream regions equally accessible. The TSS includes putative transcription factor binding sequence motifs for IKAROS, MYB, and TFAP2a as determined by PROMO and TRANSFAC prediction software. Silencing with IKAROS siRNA smart pool in HEK293 cells enhanced *ORAI3* transcription, while silencing of MYB and TFAP2a did not have any effect (Fig. 7b and Supplementary Fig. 6a–c). Silencing experiments in naive CD4$^+$ T cells confirmed the *ORAI3*-repressive function of IKAROS (Fig. 7c). Conversely, forced overexpression of IKAROS suppressed the activity of an *ORAI3* TSS reporter construct (Fig. 7d). Chromatin immunoprecipitation-polymerase chain reaction with IKAROS-specific Abs yielded signals for sequences encompassing the *ORAI3* promoter (Fig. 7e). Functionally, *IKAROS* silencing augmented the AA-induced Ca$^{2+}$ influx (Fig. 7f), ERK phosphorylation (Fig. 7g), and CD69 expression in naive CD4$^+$ T cells (Fig. 7h). Taken together, these experiments identify IKAROS as a repressive regulator of *ORAI3* transcription.

To determine how IKAROS deficiency to the extent seen in RA patients changes the transcriptome, we performed RNA-sequencing (RNA-seq) of naive CD4$^+$ T cells from eight HC transfected with control or IKAROS siRNA. Data were analyzed using a negative binomial generalized linear model after conditional quantile normalization to increase the sensitivity to identify significant changes. Surprisingly, as shown in the MA plot in Supplementary Fig. 8a, changes in the transcriptome were small, although IKAROS transcripts were clearly reduced, and involved mostly genes with low transcript numbers, among them ORAI3.

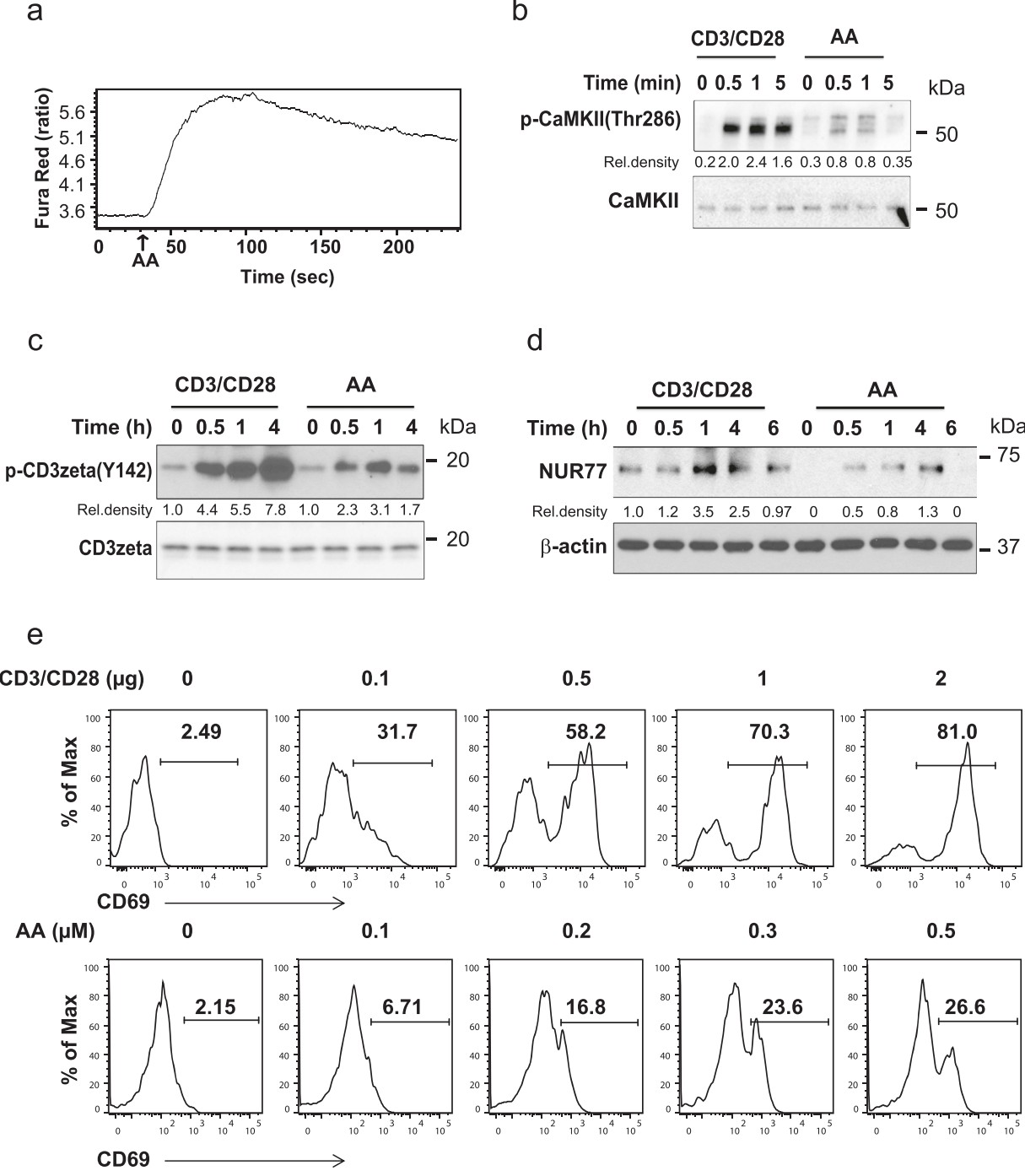

**Fig. 3 Functional consequences of arachidonic acid stimulation of naive CD4+ T cells. a** Flow cytometry analysis of AA-induced Ca$^{2+}$ influx after stimulation with 0.1 μM AA. The ratio of shifts in Fura Red fluorescence (at 406 and 532 nm) is shown at indicated times. **b** Immunoblot analysis of Ca$^{2+}$-dependent CaMKII phosphorylation in CD4+ naive T cells after crosslinking with immobilized 100 ng/ml anti-CD3/CD28 mAb or 0.3 μM AA stimulation. **c** Immunoblot analysis of CD3ζ (Y142) phosphorylation in CD4+ naive T cells after 0.3 μM AA activation or stimulation with immobilized 100 ng/ml anti-CD3/CD28 mAb. **d** Immunoblot analysis of NUR77 expression in naive CD4+ T cells after 0.3 μM AA or 100 ng/ml anti-CD3/CD28 mAb stimulation. All uncropped Western blots in Supplementary Fig. 9. **e** Flow cytometric analysis of CD69 expression in naive CD4+ T cells stimulated by increasing concentrations of AA for 3 min and then cultured for 24 h. Anti-CD3/CD28-induced CD69 expression was used as a positive control. Results in **a**–**e** are each representative of two experiments.

Only 100 differentially expressed genes were identified that are shown in the Volcano plot in Supplementary Fig. 8b. Log counts per million *z*-scores for the top 30 differentially expressed genes are shown as heat plot in Supplementary Fig. 8c. With the exception of ORAI3, none of the differentially expressed genes has an obvious role in T cell activation or effector differentiation.

**Reduced IKAROS expression in T cells from RA and PsA patients.** To determine whether IKAROS is involved in the differential regulation of *ORAI3* transcription in RA and PsA compared to normal naive CD4+ T cells, we examined chromatin accessibility and transcription of the IKAROS-encoding gene *IKZF1* using ATAC-sequencing data of isolated naive and central

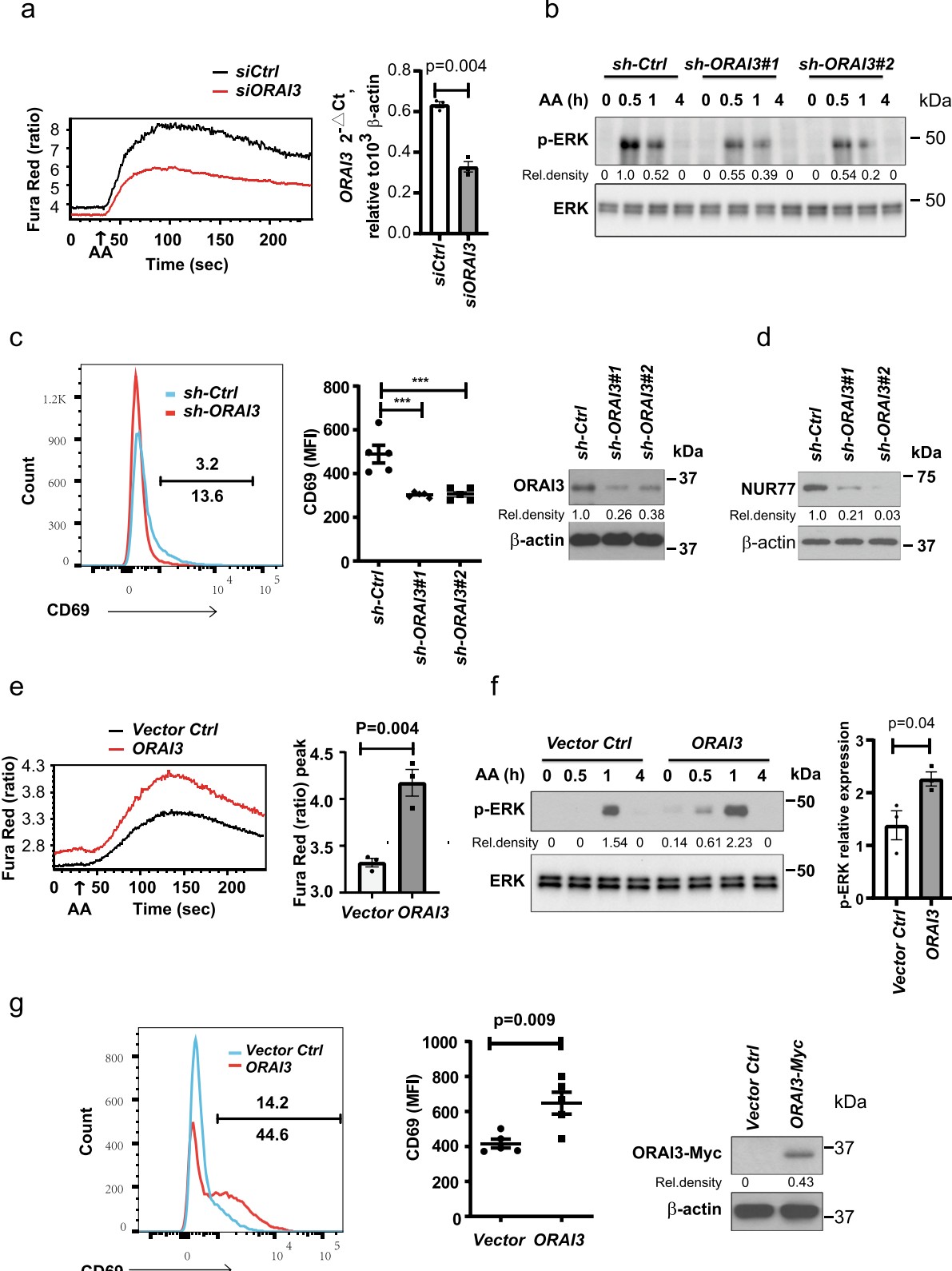

memory (CM) CD4+ T cells from nine HCs, 15 to 17 patients with RA, and 10 PsA patients. A large region including the TSS was accessible in naive and less so in CM CD4+ T cells from HC. Accessibility was significantly reduced for naive and CM CD4+ T cells from both RA and PsA patients compared to HC (Fig. 8a). Several additional more distal gene regulatory regions of *IKZF1* were also more accessible in HC T cells. *IKZF1* transcription was

quantified in naive CD4+ T cells from HC and from patients with RA, PsA, and SLE by quantitative polymerase chain reaction (qPCR) using a primer set that spans the second and third exon. *IKAROS* transcripts were significantly reduced in RA and PsA, but not SLE (Fig. 8b). Reduced expression was confirmed at the protein level by Western blotting (Fig. 8c). Immunoblots with IKAROS-specific monoclonal Ab yielded two dominant bands,

**Fig. 4 Arachidonic acid acts through ORAI3 to activate T cells. a** Naive $CD4^+$ T cells were transfected with ORAI3 siRNA smart pool or control siRNA; after 48 h, $Ca^{2+}$ influx was monitored over time (210 s) by flow cytometry of Fura Red with 0.1 μM AA added after the first 30 s (left). Right, knockdown efficiency. **b** $CD4^+$ naive T cells transduced with lentivirus expressing ORAI3 shRNA #1, ORAI3 shRNA #2, or control shRNA were assayed for p-ERK (Thr202/Tyr204) at 0.5, 1, or 4 h after 0.3 μM AA stimulation. Uncropped Western blots in Supplementary Fig. 9. **c** Representative histograms of constitutive CD69 expression in Jurkat cells stably transduced with ORAI3 shRNA or control shRNA (left). Results of CD69 expression in control and ORAI3-silenced Jurkat cells from five experiments (middle, *** = $p < 0.0001$). The efficiency of ORAI3 silencing was confirmed by immunoblotting (right). **d** Immunoblot analysis of NUR77 expression in Jurkat T cell transduced with ORAI3 shRNA #1, ORAI3 shRNA #2, or control shRNA. Data represent one of two independent experiments. **e** Flow cytometry of AA-induced $Ca^{2+}$ influx as determined from the ratio of shifts in Fura Red (at 406 and 532 nm) fluorescence in Jurkat T cells stably overexpressing *ORAI3*. Representative tracing of Fura Red ratios (left) and the peak calcium influxes from three experiments are shown (right). **f** Immunoblot of ERK phosphorylated at Thr202/Tyr204 from *ORAI3*-overexpressing Jurkat cells at indicated time points (0, 0.5, 1, and 4 h) after 0.3 μM AA stimulation (left) and peak p-ERK responses from three experiments are shown (right). **g** Flow cytometry of constitutive CD69 expression by *ORAI3*-overexpressing and vector control Jurkat T cells (left). CD69 expression summarized from five experiments (middle). Immunoblots documenting ORAI3 overexpression (right). Uncropped Western blots of **c**, **d**, **f**, **g** in Supplementary Fig. 10. Data are presented as mean ± SEM (**a**, **c**, **e**, **f**, **g**). Results in **a**, **b**, **e**, and **f** are each representative of three experiments. Data were analyzed by paired two-tailed Student's $t$ test (**a**), by one-way ANOVA followed by Tukey's multiple comparison test (**c**), or by unpaired two-tailed Student's $t$ test (**e**–**g**). Source data are provided as a Source Data file.

both of which were reduced in naive $CD4^+$ T cells from RA patients. Flow cytometric studies showed that the expression of IKAROS had a bimodal distribution, clearly distinguishing two populations of T cells that do or do not express IKAROS (Supplementary Fig. 7). The population of non-expresser or low expresser increased with differentiation. In both RA patients and HC, ~95% of naive $CD4^+$ T cells expressed IKAROS at high levels and the fraction of low expressers was small; in contrast, a larger subset of CM cells had distinctly low levels. Frequencies of the IKAROS$^{low}$ T cell subsets did not differ between HC and RA patients (Supplementary Fig. 7c). However, IKAROS$^{high}$ naive $CD4^+$ T cells from RA had a significantly lower IKAROS expression compared to those from HCs ($p = 0.007$, Supplementary Fig. 7d, unpaired two-tailed Student's $t$ test), consistent with the lower transcript numbers and the immunoblots. Expression levels in CM from RA patients were also lower; however, the difference no longer reached significance (Supplementary Fig. 7d).

**IKAROS deficiency increases synovial inflammation**. As shown in Fig. 6, ORAI3 expression accounts for the ability of RA T cells to induce synovial inflammation in the human synovium-NSG mice chimera system. To determine whether IKAROS deficiency unleashes the T cell response, NSG mice engrafted with human synovial tissue were reconstituted with CD45RO$^-$ PBMC from HC transfected with control or IKAROS siRNA. As shown in histological studies, *IKAROS* silencing increased the synovial T cell infiltrate ($p < 0.0001$, unpaired two-tailed Student's $t$ test) and the frequencies of IFN-γ-producing cells in the tissue ($p < 0.0001$, Fig. 8d, unpaired two-tailed Student's $t$ test). In parallel, synovial inflammation was assessed by gene expression analysis. Reduced IKAROS expression facilitated the synovial infiltration of differentiated effector T cells as seen from the increased expression of RORγt and TBX21, while FOXP3 levels were not different. Transcripts of T cell- and macrophage-derived mediators were increased including RANKL (*TNFSF11*), IFN-γ, IL-17A, TNF, IL-1β, and IL-6 (Fig. 8e).

## Discussion

Here, we describe that naive RA and PsA T cells exhibit increased sensitivity to respond to AA with a $Ca^{2+}$ influx due to overexpression of ORAI3, a constituent of the ARC channel. Increased *ORAI3* transcription was due to the reduced expression of the repressor IKAROS that binds to the promoter of *ORAI3*. The AA-induced $Ca^{2+}$ flux was not only sufficient to activate calcium-dependent kinases such as CAMKII but also triggered

phosphorylation of key components of the TCR signaling pathways such as CD3ζ and ERK in the absence of additional TCR or CD3 stimulation, and it induced expression of CD69 and NUR77 indicative of T cell activation. These data suggest that AA-mediated ARC activity prime the TCR signaling cascade and may account for the increased constitutive TCR signaling in circulating RA naive T cells described in Fig. 1. Such a mechanism could enable a response to low avidity self-antigens and thereby contribute to the sustained autoimmune inflammation seen in RA and PsA patients. In support of this notion, silencing of ORAI3 in adoptively transferred T cells from RA patients attenuated inflammation in the human synovium mouse chimera model, while silencing of the upstream ORAI3 repressor IKAROS enabled T cells from HC to induce synovial inflammation.

Calcium entry into cells is a key component of the TCR signaling pathway inducing the activation of the calcineurin-NFAT (nuclear factor of activated T cell) pathway and of $Ca^{2+}$-regulated kinases. The major mechanism in T cells is the store-operated mode of $Ca^{2+}$ entry[39,40]. TCR stimulation induces the production of inositol 1,4,5-triphosphate via PLCγ resulting in the depletion of $Ca^{2+}$ from ER stores and the translocation of STIM proteins to an ER plasma membrane junction[41]. STIM proteins then open the CRAC channel by binding to ORAI1. Store-independent $Ca^{2+}$ channels also function in T cells and contribute to activation. In particular, the P2X receptors have been shown to provide a costimulatory signal to T cell activation. Activated T cells secrete ATP, which binds to the P2X receptors and induce a $Ca^{2+}$ influx[42]. ARC channels are also independent of store depletion and they are activated by low concentrations of AA[16]. The $Ca^{2+}$ influx mediated by the ARC channel in T cells was small compared to that of the CRAC channel, but sufficient to induce CaMKII and CD3ζ phosphorylation. These findings are reminiscent of the function of ORAI3 in estrogen receptor-positive breast cancer cells. ORAI3 calcium channel, overexpressed in most ER$^+$ breast tumors, regulates tumor cell growth, invasion, and in vivo breast cancer progression by controlling NFAT activity and ERK1/2 and FAK phosphorylation. Moreover, ORAI3 is able to confer resistance to cell death by modulating p53 protein expression via the PI3K/Sgk-1 signaling pathway[21,24,26]. We do not know whether ORAI3-induced $Ca^{2+}$ influx was sufficient to activate the entire TCR signaling pathway, including the $Ca^{2+}$-induced translocation of NFAT; however, the finding that ORAI3 was needed for synovial inflammatory T cell responses indicates that it is at least a contributing factor.

The ARC channel is assembled of three ORAI1 and two ORAI3 subunits[43], with the ORAI3 subunit entirely responsible for the binding of AA[44]. Increased expression of ORAI3 in RA

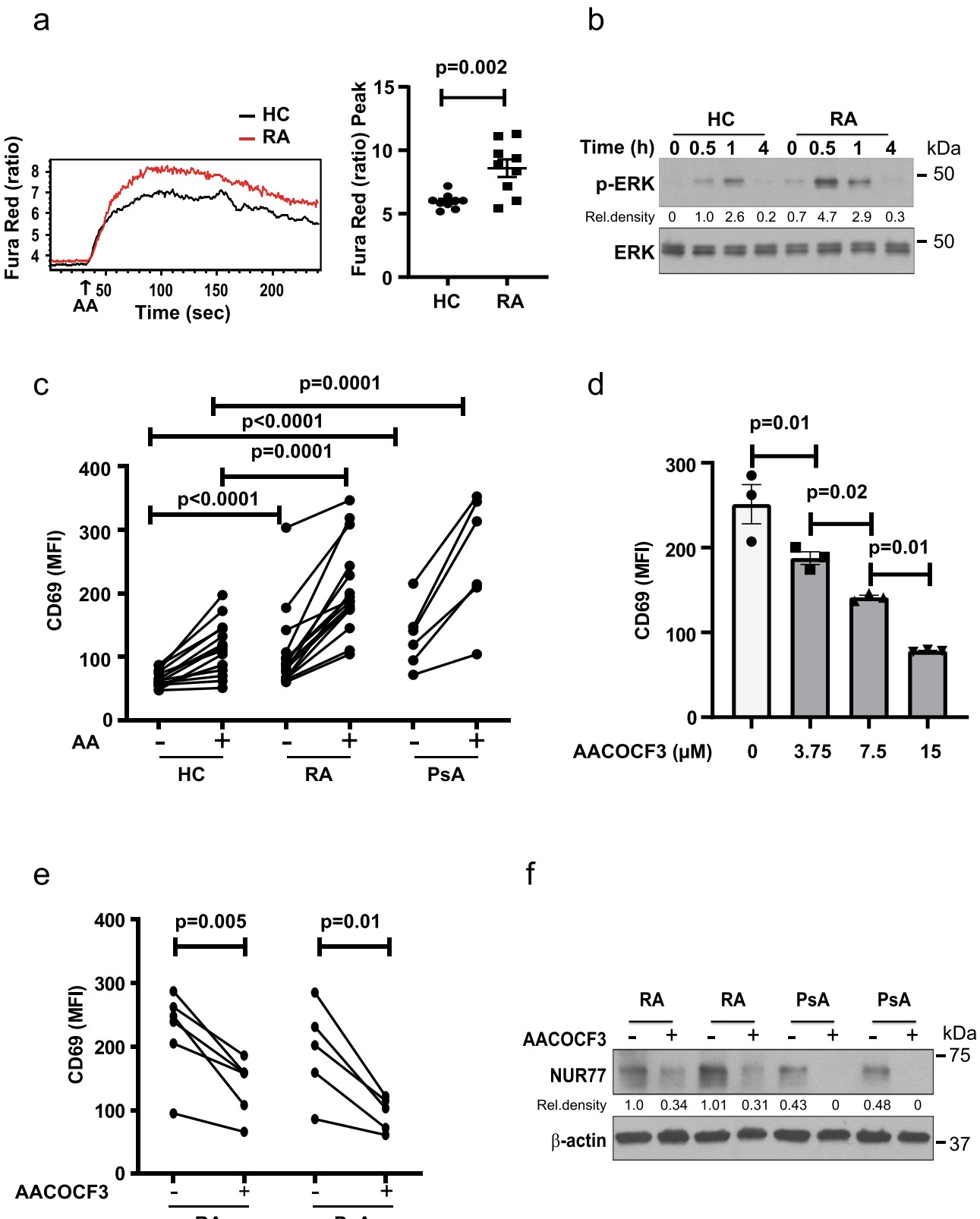

patients or forced overexpression of ORAI3 increased intracellular $Ca^{2+}$ after AA stimulation, while partial ORAI3 silencing dampened the $Ca^{2+}$ signals. We did not perform patch clamping of isolated naive $CD4^{+}$ T cells to measure ionic flux and therefore cannot make any direct conclusions on channel activity. However, the changes in intracellular $Ca^{2+}$ concentration depending on ORAI3 concentrations and AA stimulation as well as the observed increase in $Ca^{2+}$-regulated phosphorylation of CaMKII is consistent with increased channel activity. It cannot be excluded that the relative expression levels of ORAI1 and ORAI3 changes the subunit stoichiometry; however, studies by Mignen et al.[43] support the notion that the physiological ARC channel

**Fig. 5 Increased responsiveness of RA and PsA CD4$^+$ naive T cells to AA stimulation. a** Fura Red-loaded purified CD4$^+$ naive T cells were stimulated with 0.1 μM AA. Representative tracing of shifts in Fura Red fluorescence (at 406 and 532 nm) (left) and data from HC ($n = 9$) and RA ($n = 9$) patients after AA stimulation are shown as dot plots and mean ± SEM. **b** ERK phosphorylation after AA stimulation of HC and RA CD4$^+$ naive T cells. Immunoblots are representative of two experiments. Uncropped Western blots in Supplementary Fig. 10. **c** PBMC from HC ($n = 15$), RA ($n = 14$) or PsA ($n = 6$) were stimulated with 0.3 μM AA for 3 min followed by culturing for 24 h. MFI of CD69 in gated CD3$^+$CD4$^+$CD45RA$^+$CD62L$^+$ naive T cells with and without AA stimulation. **d** PBMCs were incubated with increasing concentrations of the phospholipase A2 inhibitor arachidonyl trifluoromethyl ketone (AACOCF3) to inhibit endogenous AA production; MFI of CD69 expression by naive CD4$^+$ T cells in gated CD3$^+$CD4$^+$CD45RA$^+$CD62L$^+$ was determined by flow cytometry after 24 h ($n = 3$). Data are shown as mean ± SEM. **e** PBMC from RA ($n = 6$) or PsA ($n = 5$) patients were incubated with or without 6 μM AACOCF3; CD69 expression by naïve CD4$^+$ T cells after 24 h is shown. **f** Purified naive CD4$^+$ T cells from two RA or two PsA patients were incubated with 6 μM AACOCF3 for 24 h, cells were then collected and analyzed for NUR77 protein expression. Uncropped Western blots in Supplementary Fig. 10. The immunoblot is representative of two experiments. Data were analyzed by unpaired (**a**, **c**, and **d**) or paired (**e**) two-tailed Student's *t* test. Source data are provided as a Source Data file.

exists as a pentameric structure. Since expression of ORAI1 is much higher in T cells than that of ORAI3, the effects of enhanced ORAI3 expression would then rely on increasing the number of ARC channels. ORAI1/ORAI3 heteromeric channels can be store-operated, due to ORAI1 binding STIM1[17,19,45]. However, the inclusion of more than one ORAI3 to an ORAI1 calcium channels reduces calcium ion conductance, suggesting that increased ORAI3 expression does not enhance the TCR-induced store-operated calcium entry. In support of this notion, ORAI3 silencing did not affect store-operated calcium entry upon thapsigargin stimulation. AA-induced ARC channel activity is intact even when recruited to STIM1, suggesting that CRAC and ARC channel cooperate in T cell activation[46]. A recent study has proposed that critical phosphorylation sites on CD3ζ are buried in the cytoplasmic membrane interacting with membrane phospholipids and that a Ca$^{2+}$ signal can dissociate the cytoplasmic domain of CD3 and makes it accessible to LCK (lymphocyte-specific protein tyrosine kinase)-mediated phosphorylation[47]. Endogenous production of AA and stimulation of the ARC channel may provide this Ca$^{2+}$ signal and keep the TCR signaling machinery in a metastable change. Indeed, inhibition of the endogenous AA production reduces the constitutive expression of activation markers in RA naive CD4$^+$ T cells (Fig. 5).

We identified the transcriptional repressor IKAROS as an important regulator of the *ORAI3* promoter (Fig. 7). Silencing of IKAROS increased reporter activity of an *ORAI3* promoter-reporter construct as well as increased *ORAI3* transcription in naive CD4$^+$ T cells from healthy individuals. Moreover, in correlative studies, we found an association of reduced IKAROS with increased ORAI3 expression. Naive CD4$^+$ T cells from RA had lower IKAROS than cells from healthy individuals. In addition, silencing of IKAROS reproduced the AA-induced functional changes in T cells caused by increased ORAI3 expression (Fig. 7). Most importantly, silencing of IKAROS increased the ability of T cells to induce synovitis in a human synovial tissue mouse chimera model (Fig. 8).

IKAROS encoded by *IKZF1* is a master regulator of lymphopoiesis[48]. Mice homozygous for an *IKZF1*-null mutation fail to develop T cells[49,50]. Mice engineered with a dominant-negative *IKZF1* mutation undergo rapid T cell transformation to a neoplastic state[51]. In humans, *IKZF1* haploinsufficiency results in acute lymphoblastic leukemia with a high risk of relapse[52]. While many IKAROS studies have focused on lymphocyte development, IKAROS also controls the activation of mature T cells. T cells with reduced IKAROS activity require fewer TCR engagements for proliferation and are less sensitive to inhibition of proximal signaling pathways[53]. Thus, it was surprising to find that partial silencing of IKAROS had rather minimal effects on the transcriptome beyond ORAI3 expression in nonactivated T cells.

Genome-wide association studies have implicated *IKFZ1* as a susceptibility gene in SLE supporting the notion that it can play a

role in autoimmunity[54]. Moreover, IKAROS has been implicated in the transcriptional regulation of PP2A, which is highly expressed in SLE T cells and accounts for some of their signaling abnormalities[55–57]. We, therefore, explored whether naive CD4$^+$ T cells from SLE patients have a similar increase in *ORAI3* and decrease in *IKAROS* transcripts as those from RA and PsA patients; however, we did not observe a decline in SLE T cells.

In contrast to SLE, genetic predisposition to RA and PsA has not been linked to a polymorphism of *IKZF1*. Instead, accessibility at the *IKZF1* promoter as well as several gene regulatory regions was reduced, independent of any disease-associated SNPs, supporting the notion that the reduced transcription of *IKZF1* in naive CD4$^+$ T cells is epigenetically determined and acquired. Whether reduced IKAROS expression precedes disease and therefore predisposes to disease or whether it is acquired in the course of the disease and is therefore an amplification factor remains unknown. Reduced *IKZF1* accessibility and transcription is seen with differentiation (Fig. 8); however, in both RA and PsA, reduced IKAROS expression was not limited to a small number of effector T cells that could have developed this feature as a consequence of chronic stimulation. On the contrary, decreased IKAROS and increased ORAI3 expression was a global feature of RA and PsA T cells and in particular naive CD4$^+$ T cells. T cells contribute to the pathogenesis of both RA and PsA. Reduced IKAROS and the associated increased ORAI3 expression could confer increased responsiveness to T cell stimulation irrespective of the nature of the antigen. The result of increased synovial inflammation in the human mouse chimera model after adoptive transfer of IKAROS-deficient normal T cells are supportive of this interpretation.

It remains unknown what causes the epigenetic changes leading to reduced accessibility at *IKFZ1* regulatory regions including the promoter. Both RA and PsA patients in our study were on similar treatment regimens, in particular including TNF-neutralizing medications and methotrexate. We did not see a relationship between treatment type and expression of ORAI3. Moreover, transcription factors related to the TNF pathway, such as nuclear factor-κB have not been implicated in *IKFZ1* regulation. Recent studies have mapped several enhancer regions that control *IKZF1* promoter activity[58]. Only two of these enhancers were able to stimulate transcription in T cells. One of these two enhancer regions is regulated by the transcription factor TCF1 that is important in maintaining stem-like features in naive and memory T cells and is lost with activation or effector cell differentiation[59–61]. Taken together, it appears to be unlikely that the reduced *IKZF1* transcription is a consequence of disease activity or treatment.

In summary, we have shown that IKAROS expression is reduced in naive CD4$^+$ T cells from RA and PsA patients and that reducing IKAROS expression increases the propensity of T cells to cause synovial inflammation. One of the IKAROS target genes in T cells is *ORAI3*, which, together with ORAI1, forms the

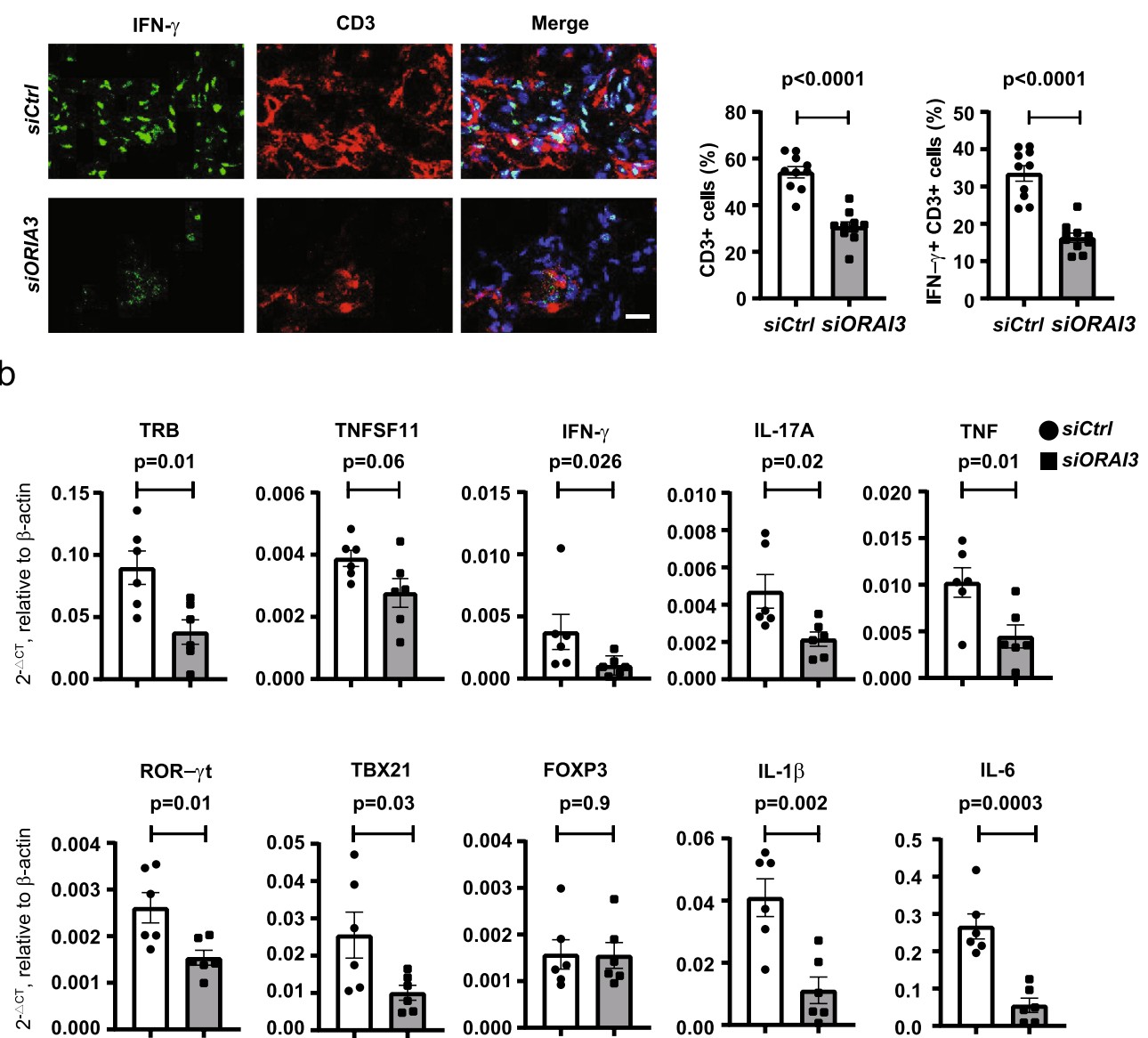

**Fig. 6 Silencing of ORAI3 in RA T cells reduces synovial inflammation. a, b** NSG mice engrafted with human synovial tissue were reconstituted with CD45RO⁻ PBMC from RA patients transfected with control or ORAI3 siRNA. Tissue was collected on day 7 after reconstitution and analyzed. **a** Three-color immunofluorescence staining of tissue sections for IFN-γ and CD3 (left). Scale bars, 20 μm. Frequencies of CD3⁺ and IFN-γ⁺CD3⁺ T cells in synovial tissues are shown as mean ± SEM of ten random fields of six synovial grafts for each treatment arm (right). Data were analyzed with unpaired two-tailed Student's *t* test. **b** Expression of genes encoding lineage-determining transcription factors and key inflammatory markers as determined by RT-PCR. Results from tissues of mice (*n* = 6) adoptively transferred with control cells are shown as white bars and circles, and those from mice with ORAI3-silenced cells as gray bars and squares. Data are presented as mean ± SEM and analyzed with unpaired two-tailed Student's *t* test. Source data are provided as a Source Data file.

ARC channel. Activation of the ARC channel by AA primes the TCR signaling pathway and may, therefore, be involved in the loss of tolerance. Interventions preventing the downregulation of IKAROS might decrease the risk of developing RA or prevent the amplification of the disease process. Alternatively, inhibition of AA production, generated by PLA2[62] or by PLC and DAG lipase, may reduce ARC channel activity and be protective.

## Methods

**Study populations**. Peripheral blood was obtained from 97 patients with RA, 15 patients with PsA, 6 patients with gout, 7 patients with SLE and 119 HCs. RA patients fulfilled the ACR diagnostic criteria for RA and were positive for rheumatoid factor and/or anti-citrullinated protein Abs. Demographic and clinical data

are given in Supplementary Table 1. RA and PsA patients were predominantly male reflecting the gender distribution at the Palo Alto Veterans Administration Rheumatology Clinic. Healthy individuals did not have a personal or family history of autoimmune diseases and no history of cancer. Chronic diseases such as hypertension or diabetes mellitus were included as long as controlled on oral medication. The protocol was approved by the Stanford University Institutional Review Board, and all participants gave written, informed consent.

**Cell purification**. PBMCs were purified from venous blood by density gradient centrifugation with Ficoll (STEMCELL Technologies, #07851). CD4⁺ naive T cells were purified from PBMC by a two-step selection procedure; first, CD4⁺ T cells were isolated with Human CD4⁺ T Cell Enrichment Cocktail (STEMCELL Technologies, #15062), then CD45RO⁺ cells were depleted to enrich for naive CD4⁺ T cells (STEMCELL Technologies, #19155). Subset purity of

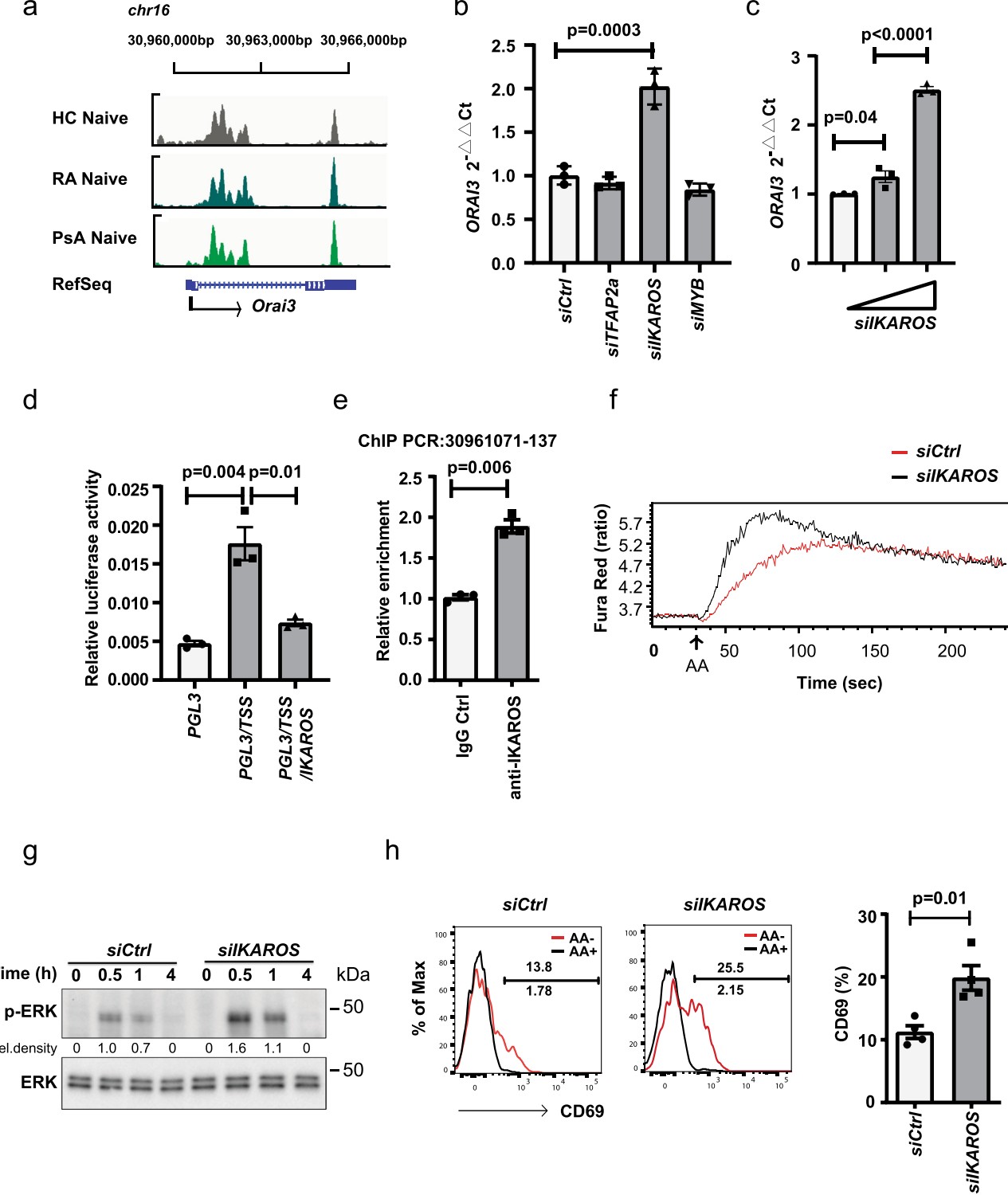

CD4[+]CD45RO[−]CD45RA[+] T cells monitored by fluorescence-activated cell sorting routinely exceeded 95%.

**Reagents and antibodies**. AA (ab120916), thapsigargin (ab120286), ORAI3 Ab (ab115558), and NUR77 Ab (ab109180) were purchased from Abcam. Abs recognizing p-CAMKII (Thr286, #12716), p-ERK (Thr202/Tyr204, #4377s), CAMKII (#4436), ERK (#9695), Myb (#12319), AP-2a (#3208), and IKAROS (#9034s) were from Cell Signaling. Abs to β-actin (#A5441) were obtained from Sigma, to p-CD3ζ (Y142) from BD Biosciences (558402) and to CD3ζ from Bio-Legend (644102). Abs for flow cytometry were purchased from BD Biosciences [CD4-V450 (560345), CD8-PE-Cy7 (557746), CD45RA-AF700 (560673), CD69-Percp-Cy5.5 (560738), p-ERK (Thr202/Try204)-AF670 (561992), and p-SLP76

(Y128)-AF647 (558438)] or BioLegend [CD62L-PE (304806), IKAROS-AF647 (368404), and CD3-AF488 (30045)]. The LIVE/DEAD[TM] Fixable Aqua Dead Cell Stain Kit from Molecular Probes, Thermo Fisher Scientific was used to gate out dead cells; Fluo-8 (21080) and Fura Red[TM] (F3021) for staining cytoplasmic Ca[2+] was from AAT Bioquest, Inc., and Thermo Fisher Scientific, respectively.

**Flow cytometry**. Freshly purified PBMCs from RA patients and HCs were stained with Abs to CD3, CD4, CD45RA, and CD62L on ice in phosphate-buffered saline (PBS) containing 1% bovine serum albumin (BSA). Patient and HC samples were always run in parallel to reduce batch effects in fluorescence intensities. Cells were washed once with 1% BSA PBS and fixed with pre-warmed BD Cytofix buffer for 10 min (BD Biosciences, #554655). Cells were pelleted and resuspended in 1 ml of

**Fig. 7 Transcriptional control of *ORAI3*. a** Chromatin accessibility maps of naive CD4+ T cells from RA and PsA patients and from healthy individuals were generated by ATAC-seq. Accessible regions at *ORAI3* TSS region (chr16:30,960,352–1215) and downstream of TSS (chr16:30,964,994–5367). **b** Accessible regions were analyzed for transcription factor (TF) motifs. Indicated TFs were silenced using siRNA transfected into HEK293T cells for 48 h, non-targeting siRNA was used as a negative control. *ORAI3* transcripts were quantified by qPCR and normalized to β-actin. Data are presented as mean ± SEM from three independent experiments ($n = 3$). Knockdown efficiency is shown in Supplementary Fig. 6. **c** Purified CD4+ T cells were transfected with increasing concentrations of IKAROS siRNA smart pool for 48 h. *ORAI3* transcription was analyzed by qPCR and normalized to β-actin. Results are triplicates from one of two independent experiments. Data are presented as mean ± SEM. **d** The sequence (860 bp) upstream of *ORAI3* TSS was cloned into the PGL3-basic plasmid and transfected into HEK293T cells alone or with an IKAROS-expressing plasmid. Firefly luciferase activity relative to Renilla luciferase is shown. Data are presented as mean ± SEM from three independent experiments ($n = 3$). **e** ChIP assays of CD4+ T cells using anti-IKAROS antibodies and a primer set amplifying the indicated potential binding site. Data are presented as mean ± SEM from three independent experiments ($n = 3$). **f** Flow cytometry analysis of AA-induced Ca$^{2+}$ influx in *IKAROS* siRNA smart pool- and control siRNA-transfected CD4+ naive T cells. Histograms are representative of two experiments. **g** Immunoblot analysis of AA-induced ERK phosphorylation in CD4+ naive T cells transfected with IKAROS siRNA smart pool; one of two experiments. Uncropped Western blots in Supplementary Fig. 10. **h** Flow cytometry analysis of AA-induced CD69 expression in purified naive CD4+ T cells transfected with IKAROS siRNA smart pool. Representative histograms and data from four experiments. Data in **b**, **c** were analyzed with one-way ANOVA followed by Tukey's multiple comparison test with adjustment. Data in **d**, **e**, and **h** were analyzed with unpaired two-tailed Student's *t* test. Source data are provided as a Source Data file.

BD Phosflow Perm buffer (#558050) for 30 min on ice. After washing twice with 1% BSA PBS, cells were incubated with Abs to p-SLP76 (Y128) or p-ERK (Thr202/Tyr204) at room temperature for 30 min in the dark. Fluorescence data were obtained on an LSRII or a Fortessa Cell Analyzer (BD Biosciences).

**Cytometric measurements of cytoplasmic Ca$^{2+}$ concentrations**. A total of $2 \times 10^6$ purified CD4+ naive T cells from RA patients or HCs were resuspended in 37 °C Hank's balanced salt solution (HBSS) with 1 μM Fura Red$^{TM}$ and 0.01% Pluronic F-127 and incubated at 37 °C for 30 min. Cells were washed with HBSS buffer once, resuspended in 1 ml HBSS, and allowed to equilibrate for 10 min in 37 °C. Fura Red fluorescence was measured by excitation with the Violet laser (406 nm) and the Green laser (532 nm). The fluorometric ratio (406 nm/532 nm) was calculated as the signal increase stimulated by the Violet laser over the signal decrease induced by the Green laser using the kinetics tool in FlowJo software version 9.3.3[63].

In the Fluo-8 cytometric experiments, PBMCs from RA patients and HCs were stained with Abs to CD3, CD4, CD45RA, and CD62L on ice in PBS containing 1% BSA. Cells were loaded with 2 μM Fluo-8 at room temperature (RT) for 20 min in HBSS buffer (Gibco, Thermo Fisher Scientific) containing both Ca$^{2+}$ and Mg$^{2+}$. Fluo-8 MFIs were recorded on an LSRII or a Fortessa Cell Analyzer (BD Biosciences) for indicated times; 0.1 μM AA was added at 30 s into the recording.

**Functional read-outs of AA stimulation**. A total of $5 \times 10^6$ naive CD4+ T cells were stimulated with 0.3 μM AA; cells were assayed for CaMKII (Thr286) phosphorylation at 0, 0.5, 1, 2, and 5 min. Culture in 10 μM Ca$^{2+}$ containing medium served as a positive control. To assess the effects of AA on T cell activation, $5 \times 10^6$ naive CD4+ T cells were treated with 0.3 μM AA for 3 min. Cells were assayed for CD3ζ and ERK phosphorylation or NUR77 expression by Western blotting at indicated times after AA stimulation. Naive CD4+ T cells stimulated on 100 ng CD3/CD28 Ab-coated plates served as a positive control. Alternatively, PBMCs ($1 \times 10^6$/ml) from HC or RA patients or Jurkat cells were stimulated with 0.3 μM AA for 3 min and analyzed by flow cytometry for CD69 expression after 24 h culture in medium without additional stimulation.

**AA measurements by enzyme-linked immunosorbent assay**. Plasma AA from RA or PsA patients or HC was quantified using the Human AA Competitive ELISA (enzyme-linked immunosorbent assay) Kit following the manufacturer's instructions (MyBioSources, #MBS703581). All samples were run in duplicates. Briefly, 50 μl of 1:50 diluted plasma was added to anti-AA Ab-coated wells, followed by the addition of 50 μl horseradish peroxidase enzyme AA conjugate. Plates were incubated for 40 min at 37 °C, washed five times and then incubated with 90 μl 3,3′,5,5′-tetramethylbenzidine substrate per well for an additional 20 min at 37 °C. Reactions were stopped with 50 μl stop solution, and plates were read at 450 nm within 5 min. Concentrations were calculated with the CurveExpert 1.3 professional software.

**ATAC-seq**. CD4 T cells were isolated using Human T cell Enrichment Cocktail (STEMCELL Technologies, Canada), followed by purification of naive (CD3+CD4+CD62L+CD45RA+CD28+), CM (CD3+CD4+CD62L+CD45RA−CD28+), and effector memory (CD3+CD4+CD62L−CD45RA−CD28+) subsets by fluorescence-activated cell sorting. ATAC-seq libraries were generated using 50,000 cells for each subset. Cells were first washed with cold PBS and RSB buffer (10 mM Tris-HCl (pH 7.4), 10 mM NaCl, 3 mM MgCl₂), followed by washing with RSB buffer containing 0.1% NP-40 and 0.1% Tween-20. Subsequently, cell pellets were resuspended in a transposase reaction mix (25 μl 2× TD buffer, 2.5 μl transposase (Illumina), and 22.5 μl nuclease-free water) and incubated at 37 °C for 30 min. DNA fragments were purified using Qiagen MiniElute Kit and the library was amplified with Nextera PCR primers. Library quality was checked on bioanalyzer and sequenced on

Illumina NextSeq 500. The sequencing reads were processed by trimming adapters and low-quality reads using in-house scripts and aligned to human reference genome hg19. Peaks were identified for each sample using macs2 and a consensus peak set was determined consisting of peaks present in at least three samples. Reads were converted into bigwig format for visualization of the genomic tracks, which were normalized to total reads mapped within the consensus peak set[64].

**RNA-seq**. Naive CD4+ T cells were isolated from PBMC using EasySep human Naive CD4+ T cell Enrichment Kit (STEMCELL Technologies, #19555). Two million cells were transfected with IKAROS siRNA smart pool (Dharmacon: L-019092-02-0005) or control siRNA (Dharmacon: D-001810-10-05) using P3 Primary cell 4D-Nucleofector Kit and the Lonza 4D-Nucleofector System (Lonza). Transfected cells were cultured for 48 h. Libraries were generated as recently described[65] and sequenced using the Novogene Hiseq platform (Novogene). The fastq files generated from the sequencing runs were analyzed using the nf-core pipeline[66] to determine read counts mapped to genes in GRCh37 genome. The data were further analyzed using Bioconductor packages edgeR and conditional quantile normalization. The experimental design was modeled upon donor and condition (control/knockdown) (~donor + condition). The downstream analysis to identify differentially expressed genes was performed as described in Chen et al.[67] with the addition of offsets from conditional quantile normalization[68], followed by the application of gene-wise negative binomial generalized linear models[69].

**Reporter gene assays**. The region upstream of the TSS of *ORAI3* (chr16:30,960,352–30,961,215) was amplified from T cell-derived genomic DNA using the primer pair 5′-GGGGGTACCCCAAGTTGTTTTATATTTCCATG-3′ and 5′-GGGCTCGAGTTAGCAAGCTCCATGAAGTCAAG-3′. PCR products were purified and cloned into the PGL3-basic plasmid (Promega). Sequences of insertions were authenticated. Thirty nanograms of PGL3-basic control, *ORAI3-TSS/PGL3* or *ORAI3-TSS/PGL3* plus 0.5 μg pcDNA-*IKFZ1* (Genescript: clone Hu28071D) together with 1 ng Renilla luciferase reporter pR-TK plasmids were cotransfected into HEK293 using Fugene HD transfection reagent (Promega, E2311). Cells were cultured for 48 h before being analyzed using the dual-luciferase reporter assay system (Promega, E1910).

**ChIP assay**. ChIP assays were performed on five million CD4+ T cells by using the ChIP-IT Kit (53040) from Active Motif. Oligonucleotide primers were designed to amplify the *ORAI3* promoter sequence (chr16/30,961,071–137; forward: 5′-TTGCTG TTATTCTGTGGTGAG-3′ and reverse: 5′-CAAAATAAGGGATCCATCAGA-3′). Normal IgG was used as control.

**Lentiviral transduction**. GFP-expressing lentiviral vectors expressing ORAI3-targeting shRNA (*pGFP-lenti-Sh-ORAI3*; OriGene: TL307587) and full-length *ORAI3*-expressing pCMV6-AC lentiviral vectors (OriGene: RC202325L1 NM_152288), respectively were packaged into HEK293T cells using the lenti-vpak lentiviral packaging kit (OriGene TR30037). Scrambled control shRNA and pCMV6-AC empty vector were used as negative controls. Jurkat cell lines stably expressing sh-*ORAI3* or *ORAI3* were established as per the manufacturer's instructions (OriGene).

**Transfection with siRNA**. Two million CD4+ naive T cells were transfected with control siRNA, ORAI3 siRNA smart pool (Dharmacon: L-015896-00), or IKAROS siRNA smart pool (Dharmacon: L-019092-02) using the P3 Primary cell 4D-Nucleofector Kit and the Lonza 4D-Nucleofector System (Lonza). Transfected cells were cultured in RPMI-1640 media supplemented with 10% FBS for 48 h before use. For screening TFs regulating ORAI3 expression, control siRNA, TFAP2a

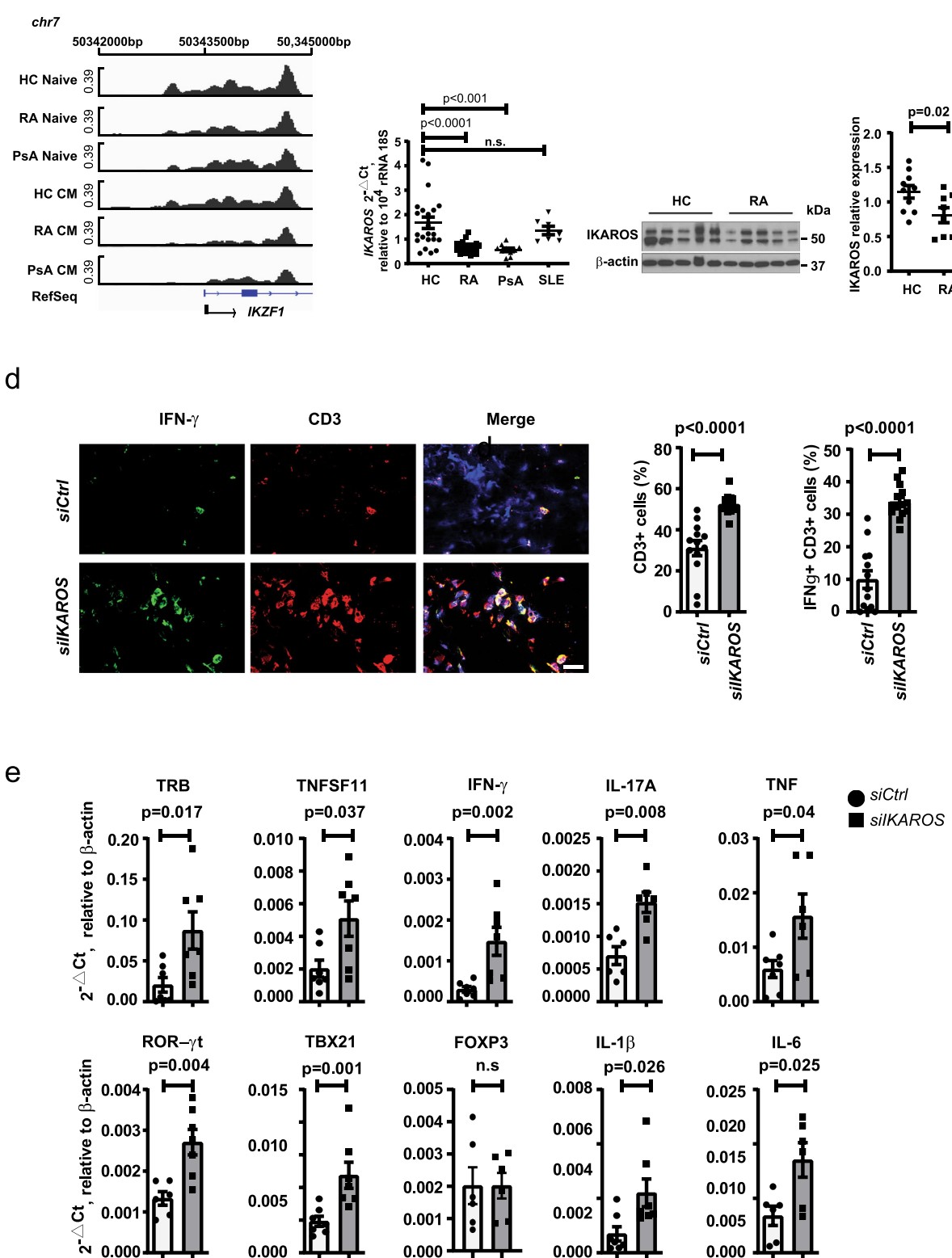

siRNA (A-006348-14), IKAROS siRNA (L-019092-02), or MYB siRNA (M-003910-00) were transfected into HEK293T cells with Fugene HD transfection reagent (Promega, E2311).

**qPCR**. Total RNA was extracted with the RNeasy plus Mini Kit (Qiagen), cDNA was synthesized with the High Capacity RNA-to-cDNA Kit (Applied Biosystems, 4387406). qPCR was performed in duplicates in 384-well plates using the ABI 7900HT System with Taqman Universal Master Mix II (Thermo Fisher, 4440040)

and the following Taqman gene expression primers: *ORAI3* (hS00743683-s1), 18S rRNA (Hs99999901_s1), *IKAROS* (HS00958474_m1 *IKZF1*), and *β–actin* (HS99999903_m1). For all other transcript analysis, qPCR was carried out using PowerUp™ SYBR™ Green Master Mix (Thermo Fisher, A25776) with primers listed in Supplementary Table 2.

**Immunoblotting**. CD4$^+$ naive T cells ($2 \times 10^6$) were lysed on ice with RIPA buffer (Thermo Scientific) supplemented with phosphatase (sodium orthovanadate) and

**Fig. 8 Reduced IKAROS expression in RA CD4+ T cells promotes synovial inflammation. a** Chromatin accessibility maps of naive and central memory (CM) CD4+ T cells from RA patients, PsA patients, and healthy individuals were generated by ATAC-seq. Accessible regions at TSS region (chr7:50,342,000–5000) of *IKFZ1* encoding IKAROS. **b** *IKFZ1* expression in purified CD4+ naive T cells from HC ($n = 22$), RA ($n = 21$), PsA ($n = 10$), and SLE ($n = 7$) patients was quantified by qPCR. Data are presented as dot plots with mean ± SEM. **c** Representative immunoblots of IKAROS in CD4+ naive T cells from HC and RA patients (left); dot plot of IKAROS intensities normalized to β-actin from HCs ($n = 10$) and RA ($n = 8$) (right). Data are presented as dot plots with mean ± SEM. Uncropped Western blots in Supplementary Fig. 10. **d, e** NSG mice engrafted with human synovial tissue were reconstituted with HC CD45RO− PBMC transfected with control or IKAROS siRNA. Tissue was collected on day 7 after reconstitution and analyzed. **d** Three-color immunofluorescence staining of tissue sections for IFN-γ and CD3 (left). Scale bars, 20 μm. Frequencies of CD3+ and IFN-γ+CD3+ T cells in synovial tissues are shown as mean ± SEM of 12 synovial grafts for each treatment arm pooled from 6 mice (right). **e** Expression of genes encoding lineage-determining transcription factors and key inflammatory markers as determined by RT-PCR. Results from tissues of mice adoptively transferred with control cells are shown as white bars and circles, those from mice with IKAROS-silenced cells as gray bars and squares. Data are presented as dot plots with mean ± SEM. Data in **b** were analyzed by one-way ANOVA followed by Tukey's multiple comparison test; data in **c**–**e** were analyzed by unpaired two-tailed Student's *t* test. n.s. not significant. Source data are provided as a Source Data file.

proteinase inhibitor cocktails (Roche, 33576300) and phenylmethylsulfonyl fluoride (Santa Cruz Biotechnology). Total protein (10 μg) was separated on 4–15% gradient SDS gels and then transferred to polyvinylidene difluoride membranes. ORAI3, NUR77, and IKAROS were detected with the respective Abs. Membranes were stripped with stripping buffer (Invitrogen) and re-probed with anti-β-actin Abs. For TCR signaling experiments, CD4+ naive T cells or Jurkat cells stably transduced with ORAI3 shRNA, control shRNA, or *ORAI3* were stimulated with AA or on immobilized 100 ng anti-CD3/CD28 Abs, lysed at indicated time points, and probed for p-CD3ζ (Y142), CD3, p-CAMKII, CAMKII, p-ERK (Thr202/Tyr204), ERK, and NUR77 by immunoblotting.

**Flow cytometry of transcription factor expression**. One million T cells were collected and washed with 2% BSA PBS and stained with Aqua, anti-CD4, anti-CD45RA, and anti-CD62L for 30 min on ice. Cells were washed once and stained for IKAROS as recommended by the manufacturer (BioLegend, #424401). In brief, cells were fixed with 1 ml Transcription Factor 1× Fix solution for 30 min at RT, 2 ml of the Transcription Factor 1× Perm Buffer was added, centrifuged at 300–400 × *g* at room temperature for 5 min and wash again with 2 ml 1× Perm Buffer. Cell pellets were resuspended in 100 μl of the Transcription Factor 1× Perm Buffer; Alexa Fluor® 647-labeled anti-human IKAROS Ab (1:50 dilution, BioLegend, #368404) was added and stained for 30 min at RT.

**Immunohistochemical staining**. Tissues were snap frozen in OCT and stored at −80 °C. Five micromole sections were air-dried and fixed with acetone at 4 °C for 10 min for staining or shock frozen for RNA extraction. Synovial T cells were identified by immunohistochemistry with mouse anti-human CD3 Abs (1:50), and IFN-γ production was detected with rabbit anti-human IFN-γ Abs (1:100) as described[38]. Sections were analyzed with an Olympus BX41 microscope and CellSense software.

**Human synovial tissue-NSG mice chimera**. NSG male mice (Jackson Laboratory, Bar Harbor, ME) were kept in pathogen-free facilities and used at the age of 8–12 weeks as described[36]. Pieces of human synovial tissue were placed into a subcutaneous pocket. On day 7 after engraftment, mice were reconstituted with 20 million CD45RO− PBMCs derived from RA patients transfected with 20 μM ORAI3 or control siRNA or HDs transfected with 20 μM IKAROS or control siRNA. All experiments were approved by the Palo Alto Veterans Administration Healthcare System Animal Care and Use Committee.

**Statistical analysis**. Statistical analysis was performed using PRISM 8.4.1 (GraphPad Software Inc., La Jolla, CA). Results are given as mean ± SEM. Unpaired two-tailed and paired two-tailed Student's *t* test or one-way analysis of variance, followed by Tukey's multiple comparison test were used as appropriate. $P < 0.05$ was considered significant.

**Reporting summary**. Further information on research design is available in the Nature Research Reporting Summary linked to this article.

## Data availability
RNA-seq and ATAC-seq data have been deposited in SRA under the accession codes PRJNA681877 and PRJNA686153. The authors declare that the other data supporting the findings of this study are available within the article and its Supplementary information files, or are available upon reasonable request to the corresponding author. Source data are provided with this paper.

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

## Acknowledgements

This work was supported by the National Institutes of Health (R01 AR042527, R01 HL117913, R01 AI108906, R01 HL142068, and P01 HL129941 to C.M.W. and R01 AI108891, R01 AG045779, U19 AI057266, and R01 AI129191 to J.J.G.), the Veterans Administration I01 BX001669 to J.J.G and the Praespero Foundation. The content is solely the responsibility of the authors and does not necessarily represent the official views of the National Institutes of Health.

## Author contributions

Z.Y., C.M.W., and J.J.G. designed research and analyzed data. Z.Y., Y.S., K.J., and J.Q. performed the experimental work. B.H. and R.R.J. performed and analyzed ATAC-seq, R.R.J. analyzed RNA-seq data. K.S. and J.J.G. recruited patients and analyzed clinical data. Z.Y., C.M.W., and J.J.G. wrote the manuscript.

## Competing interests

The authors declare no competing interests.
