## [Peer Review File · Nature Communications]

Reviewers' comments:

Reviewer #1 (Remarks to the Author):

I found this paper very clear, straightforward proving elegantly that arachidonic acid, and interestingly not CD3 engagement, causes increased calcium response and phosphorylation of downstream molecules. The authors attribute the increased expression of ORAI3 to decreased expression of IKAROS.

the authors report ATAC data on ORAI3 and IKAROS. It would be interesting to report on more global differences between normal and RA/PsA (low IKAROS expressing cells) data. IKAROS controls many genes and some insight on the global events would be interesting at this point.

Reviewer #2 (Remarks to the Author):

Ye and colleagues make the observation that CD4 T cells from patients with rheumatoid and psoriatic arthritis have heightened expression of ORAI3, arachidonic acid (AA)-regulated calcium (ARC) channel component, compared to controls, with increased calcium influx and TCR signaling upon AA exposure. ORAI3 expression is repressed by IKAROS, expression of which compared to controls was decreased in RA and PsA CD4 T cells, presumably accounting for the increased activation phenotype of the latter cells. Forced reduction of T cell IKAROS expression in a humanized mouse synovitis model resulted in enhanced inflammation. Based upon their findings, the authors conclude the aberrant regulation of IKAROS – ARC channel could lead to loss of T cell tolerance in RA and PsA.

These findings are novel, with clear supporting data. A few observations and suggestions:

It is important to show flow cytometry data, and how naïve and Tcm cells from patients with inflammatory arthritis and controls, are gated, particularly to rule out CD45RA+ cells that have been activated in vivo and then re-expressed this isoform. CCR7 staining would help confirm that the naïve cells are indeed naïve. Showing the two dimensional flow plots for the initial studies on naïve cells, and the later ones on Tcm cells, even in supplemental data, is absolutely critical. In a like manner, it is likely naïve T cells in RA are exposed in vivo to cytokines at concentrations not observed in healthy donors, and not of the same type as in SLE. Does pre-exposure of naïve cells from HC lead to the same phenotype observed in RA T cells?

The lack of relationship of the RA T cell phenotype to drug therapy and disease activity indicates that the IKAROA-ORAI3 pathway is T cell intrinsic. The authors at least initially argue that this defect is acquired, but then offer data to rebut this idea. Re: acquired phenotype, does exposure of naïve healthy cells to RA serum recapitulate the RA phenotype? Re: T cell intrinsic effects, are there RA SNPs associated with enhancers/superenhancers that control IKZF1 transcription?

It's not clear why AA-induced Ca⁺⁺ flux enhances TCR activation? Mechanistic insights? The discussion on this point does not offer much insight into mechanism. Maybe this is beyond the scope of the paper, but it a curious finding. Is this possibly an AA dose effect? It would be important in the AA triggering experiments to assess a dose range.

The finding that RA patients and HC had essentially numbers of IKAROS high naïve CD4 T cells is curious, with the IKAROS driven phenotype in RA explained by the idea that the degree of its expression in high RA cells was less than its expression in high HC cells. This is a critical point, one critical to the conclusions of the work, and one in which 2-D flow plots need to be shown as primary data, allowing the reader to see how the gating was done, with inclusion markers such as CCR7.

Reviewer #3 (Remarks to the Author):

The study by Ye et al. attempts to identify the role of Orai3 and arachidonic acid (AA) mediated calcium entry on naïve T cell function. Orai3 is a homolog of the ORAI1 channels that form the plasma membrane calcium channel canonically activated by the endoplasmic reticulum (ER) transmembrane protein STIM upon ER calcium depletion. Orai3 has also been demonstrated to be activated independently of ER depletion, by AA and its metabolites such as LTC₄. The authors observed Orai3 mRNA and protein expression are increased in naïve T cells isolated from patients with rheumatoid arthritis. They also show AA through Orai3 stimulates activation of signaling molecules that are important for T-cell receptor (TCR)-mediated signaling. The authors suggest that the transcription factor IKAROS negatively regulate Orai3 expression. There are several concerns with this manuscript and these are described in the following comments:

Major Comments

- The role of Orai3 in mediating AA-induced calcium signal needs to be more rigorously studied. Many of the experiments rely on the dye Fluo-8 and flow cytometry. The dye and methodology are prone to artifacts (i.e. changes in membrane potential, changes in dye loading, etc.). The authors should employ single cell calcium imaging with the ratiometric dyes such as Fura2. The gold standard is to study the AA-induced currents using whole cell patch clamp electrophysiology. In the absence of such evidence, no conclusions can be reached regarding the nature of the calcium entry pathway the authors are reporting.
- The western blots presented are of poor quality and not convincing. There are no molecular weights. Whole blots are not provided. Quantification is not provided. Antibodies that measure phosphorylated proteins (i.e. ERK, CD3, CAMKII) should be normalized to total amounts of those respective proteins. Multiple shRNA or siRNA sequences should be employed to rule out off-target knockdown. Blots should be provided to document knockdown (i.e. siTFA2a, siIKAROS, siMTB).
- It would be important to know the outcomes of the authors' proposed model on effector T cell function. How does aberrant ORAI3-mediated ARC signaling in naïve T cells translate to T cell development and hence effector T cell function? A mouse model of rheumatoid arthritis would greatly enhance this study. The authors can then measure such outcomes. It would also rule out any confounding factors from the human cell lines. It would be even better if there is a transgenic mouse model.

Minor Comments

- Data points should be represented individually for box plots.
- A few axes are missing from flow diagrams.
- More discussion of ORAI and calcium signaling should be included in the introduction and credit should be given to authors who described the function of Orai3 in various cell systems. The discussion should include reports describing Orai3 as a purely store-operated channel (e.g. enamel cells; breast cancer cells) as well as reports describing Orai3 and Orai1 heteromers functioning as a store independent channel.

We thank the reviewers for their helpful comments and appreciate your editorial guidance on revising the manuscript. We would like to resubmit a revised manuscript, in which Fig. 1c, Fig. 3a, Fig. 4a, 4b, 4c,4e, Fig. 5a, b Fig. 6f, g, Fig. 8 c, d and e and supplementary Fig. 1a,1b, supplementary Fig. 2, supplementary Fig.5, supplementary Fig. 6a, supplementary Fig.7 and supplementary Fig.8 are new.

In the revised manuscript, we have addressed the comments as follows:

Reviewer #1

The authors report ATAC data on ORAI3 and IKAROS. It would be interesting to report on more global differences between normal and RA/PsA (low IKAROS expressing cells) data. IKAROS controls many genes and some insight on the global events would be interesting at this point.

We appreciate the encouraging feedback from the reviewer. We also expected that the reduced expression of IKAROS, although modest, would induce widespread transcriptome changes in CD4 T cells. To address this point, we performed RNA-seq in CD4 naïve T cells comparing control-transfected or IKAROS partially silenced. Surprisingly, we found very limited changes although the study was sufficiently powered (n=8), IKAROS was appropriately partially repressed and ORAI3 was consistently upregulated (see MA plot in Figure 8c), but all other genes bordered on significance. Of note, none of the other genes that showed a trend towards differential expression, is known to be involved in T cell differentiation and the expression of inflammatory genes that are expressed in the chimera synovitis.

Reviewer #2

These findings are novel, with clear supporting data. A few observations and suggestions:

It is important to show flow cytometry data, and how naïve and Tcm cells from patients with inflammatory arthritis and controls, are gated particularly to rule out CD45RA+ cells that have been activated in vivo and then re-expressed this isoform. CCR7 staining would help confirm that the naïve cells are indeed naïve. Showing the two dimensional flow plots for the initial studies on naïve cells, and the later ones on Tcm cells, even in supplemental data, is absolutely critical.

We defined naïve CD4 T cells as co-expressing CD62L and CD45RA and therefore excluded TEMRAs that are very high in CD45RA expression. In contrast to CD8 T cells, stem-like memory CD4 T cells can be clearly distinguished from naïve CD4 T cells. Gating strategies for the different experiments is shown in supplementary Figure 1a and supplementary Figure 6a.

In a like manner, it is likely naïve T cells in RA are exposed in vivo to cytokines at concentrations not observed in healthy donors, and not of the same type as in SLE. Does pre-exposure of naïve cells from HC lead to the same phenotype observed in RA T cells.

Our studies focused on naïve cells that are not in the tissue infiltrate, but could be exposed in the circulation. We cultured naïve CD4 T cells with serum of HC and RA patients and screened cells for the expression of the activation markers found in RA patients on circulating naïve cells. We did not find a significant difference in the expression of CD69, pSLP76 and pERK between the two different treatments (supplementary figure 2).

The lack of relationship of the RA T cell phenotype to drug therapy and disease activity indicates that the IKAROA-ORAL3 pathway is T cell intrinsic. The authors at least initially argue that this defect is

acquired, but then offer data to rebut this idea. Re: acquired phenotype, does exposure of naïve healthy cells to RA serum recapitulate the RA phenotype? Re: T cell intrinsic effects, are there RA SNPs associated with enhancers/superenhancers that control IKZF1 transcription?

As mentioned above, we did not induce the phenotype observed in RA patients after exposure of healthy CD4 naïve T cells to RA serum.

We had discussed in the original manuscript that SLE has been associated with *IKFZI*-associated SNPs but not RA. Our ATAC-seq show decreased chromatin accessibility at the *IKFZI* promoter as well as several additional *IKFZI* regulatory regions, which do not contain an RA-associated SNP. We have revised the discussion to be more specific.

It's not clear why AA-induced Ca⁺⁺ flux enhances TCR activation? Mechanistic insights? The discussion on this point does not offer much insight into mechanism. Maybe this is beyond the scope of the paper, but it a curious finding. Is this possibly an AA dose effect? It would be important in the AA triggering experiments to assess a dose range.

We had shown in the original manuscript, Figure 3e that CD69 expression increased with AA concentrations. We had also shown that the AA-induced calcium influx is sufficient to induce the phosphorylation of CaMKII (Figure 3b). How calcium influx leads to the phosphorylation of other, less calcium regulated molecules such as ERK remains unclear, however, similar observations have been made for the oncogene function of ORAI3 in breast cancer cells, suggesting the activity of calcium-dependent kinase. As a second possibility, a recent paper in Nature in 2013 may point at a mechanism. In this paper, Shi XS et al have reported that an initial Ca²⁺ influx is needed to neutralize the negatively charged phospholipid in the plasma membrane, which then releases the buried tyrosine of CD3ε/ζ from the cell membrane to progress with TCR signaling. We have included a discussion on these possible mechanisms.

The finding that RA patients and HC had essentially numbers of IKAROS high naïve CD4 T cells is curious, with the IKAROS driven phenotype in RA explained by the idea that the degree of its expression in high RA cells was less than its expression in high HC cells. This is a critical point, one critical to the conclusions of the work, and one in which 2-D flow plots need to be shown as primary data, allowing the reader to see how the gating was done, with inclusion markers such as CCR7.

We have included the gating strategy as supplementary Figure 6a.

Reviewer #3

The study by Ye et al. attempts to identify the role of Orai3 and arachidonic acid (AA) mediated calcium entry on naïve T cell function. Orai3 is a homolog of the ORAI1 channels that form the plasma membrane calcium channel canonically activated by the endoplasmic reticulum (ER) transmembrane protein STIM upon ER calcium depletion. Orai3 has also been demonstrated to be activated independently of ER depletion, by AA and its metabolites such as LTC₄. The authors observed Orai3 mRNA and protein expression are increased in naïve T cells isolated from patients with rheumatoid arthritis. They also show AA through Orai3 stimulates activation of signaling molecules that are important for T-cell receptor (TCR)-mediated signaling. The authors suggest that the transcription factor IKAROS negatively regulate

Orai3 expression. There are several concerns with this manuscript and these are described in the following comments:

Major Comments

• *The role of Orai3 in mediating AA-induced calcium signal needs to be more rigorously studied. Many of the experiments rely on the dye Fluo-8 and flow cytometry. The dye and methodology are prone to artifacts (i.e. changes in membrane potential, changes in dye loading, etc.). The authors should employ single cell calcium imaging with the ratiometric dyes such as Fura2. The gold standard is to study the AA-induced currents using whole cell patch clamp electrophysiology. In the absence of such evidence, no conclusions can be reached regarding the nature of the calcium entry pathway the authors are reporting.*

We have repeated all experiments using the ratiometric dye Fura Red. New results are shown in Fig. 1c; Fig.3a; Fig.4a, e; Fig. 5a; Fig. 6f and confirm the previous findings. We agree that patch clamping is the golden standard to examine calcium flux. However, patch clamping is hardly possible in population studies.

• The western blots presented are of poor quality and not convincing. There are no molecular weights. Whole blots are not provided. Quantification is not provided. Antibodies that measure phosphorylated proteins (i.e. ERK, CD3, CAMKII) should be normalized to total amounts of those respective proteins. Multiple shRNA or siRNA sequences should be employed to rule out off-target knockdown. Blots should be provided to document knockdown (i.e. siTFA2a, siiKAROS, siMTB)

We apologize that the presentation of our Western blots gave the impression of poor quality. We have edited the images. Molecular weight markers were added to all blot images. Quantification is now shown for all blots in Fig. 3b, c, d; Fig. 4b, c, d, f; Fig. 5b, f; Fig. 6g. Phosphorylated proteins were normalized to the respective total proteins. Original full length blots are included as supplementary figures 7 and 8.

As for the siRNA for Orai3 and IKAROS, we use ORAI3 smart pool composed of 4 oligoes with 4 different targeting sequence. As for shRNA, we used two different targeting shRNA as shown in Fig. 4b-d. Knockdown efficiency for siAP2a, siIKAROS and siMyb are now shown in supplementary figure 4.

• *It would be important to know the outcomes of the authors' proposed model on effector T cell function. How does aberrant ORAI3-mediated ARC signaling in naïve T cells translate to T cell development and hence effector T cell function. A mouse model of rheumatoid arthritis would greatly enhance this study. The authors can then measure such outcomes. It would also rule out any confounding factors from the human cell lines. It would be even better if there is a transgenic mouse model.*

We prefer the human tissue mouse chimera model with IKAROS-silenced adoptively transferred T cells as shown in Figure 8 rather than a mouse ORAI3 transgenic arthritis model. Currently available mouse models only reflect limited aspects of the human disease and are not suitable to study preclinical risk factors, as we are proposing for reduced IKAROS expression.

1. Most of the arthritis model such as the collagen-induced arthritis allow to study the effector mechanisms after some artificial induction of a T cell response, usually facilitated by adjuvanted immunization and therefore do not really allow to examine the role of the ARC channel in T cell activation.
2. The K/BxN model is antibody mediated.
3. The SKG model has a defect in TCR signaling.

In contrast, the human tissue mouse chimera model allows to tailor the condition in normal human CD4 T cells to match the findings seen in patients. These so-modified T cells are then adoptively transferred to test their ability to induce disease. It should be noted that only fresh T cells were used and not human cell lines. The model has been used in numerous studies and has been accepted as one model by the scientific community.

Minor Comments

- *Data points should be represented individually for box plots.*

All box plots were changed to dot blots to see individual data points

- *A few axes are missing from flow diagrams.*

We apologized for this oversight. The figures were corrected.

- *More discussion of ORAI and calcium signaling should be included in the introduction and credit should be given to authors who described the function of Orai3 in various cell systems. The discussion should include reports describing Orai3 as a purely store-operated channel (e.g. enamel cells; breast cancer cells) as well as reports describing Orai3 and Orai1 heteromers functioning as a store independent channel.*

We have expanded the sections on ORAI 1 and ORAI 3 in the introduction and the discussion and have included a discussion on ORAI3 expression and function in breast cancer cells the function of ORAI1/ORAI3 heteromers in store dependent and independent signaling. We have added references. As far as we can see, ORAI1 and 2, but not 3 is involved in enamel.

We thank the reviewers, who have helped us improving our manuscript, and we are submitting the revised manuscript for your consideration.

REVIEWER COMMENTS

Reviewer #1 (Remarks to the Author):

i appreciate your diligence in responding to the comments of the reviewers.

Reviewer #2 (Remarks to the Author):

The authors have addressed my concerns.

Reviewer #3 (Remarks to the Author):

Figure 1

1. Include fura traces for Fig. 1C and flow trace for Fig. 1D

Figure 2

- Labeling y-axis of Figure 2A is incorrect, should just be relative expression

Figure 3

- In the text it's stated that stimulation with AA causes rapid calcium influx, and that this peak response was smaller than TCR mediated calcium influx. There are no graphs or data to show this (especially in the phosphorylation of CAMKII).
- In Fig. 3B, it's unclear how the stimulation with calcium is being performed. Are stores being depleted prior or how is just adding calcium causing phosphorylation of CAMKII?
- In Fig. 3E, it would be useful to have a positive control for CD69 expression like CD3 stimulation to know how much CD69 is upregulated in response to TCR stimulation.

Figure 4

- In Fig 4A, it would be critical to know if siOrai3 also reduces SOCE or if siOrai1 reduces AA induced calcium influx
- In Figure 4B, AA stimulation is shown to induce activation of downstream signaling molecules like ERK. Do the authors think AA stimulation can induce activation of NFAT isoforms?
- Quantifications missing for Fig. 4E and 4F.

Figure 5

- Include fura trace for Fig. 5A
- It's stated that RA CD4 naïve T cells have higher basal CD69 expression compared to healthy controls, but no statistics are shown for this (only comparisons for after AA stimulation). Looking at the baseline levels for HC and RA, they don't appear much different.
- No error bars are included for 5D. Was this a single experiment?
- Baseline levels of CD69 for RA samples in Fig. 5E are much higher than RA samples in Fig 5C. Reason for this discrepancy?

Figure 6

- Why were the siRNA experiments for IKAROS, MYB, and TFAP2a performed in HEK293 and not a more relevant model like Jurkat?
- Fig. 6H is not very convincing for increase in CD69 expression with IKAROS knockdown.

Figure 8

- One of the biggest oversights was the lack of inclusion of PBMCs transfected with siOrai3 in the NSG mouse model.
- I'm not really sure what the new sequencing data really adds in Figure 8. If anything it shows that the negative regulatory role of IKAROS is relatively weak on Orai3 expression.

Reviewer #3

Figure 1

1. Include fura traces for Fig. 1C and flow trace for Fig. 1D

Fura Red traces have been added in Fig.1c, and flow traces of CD69 expression have been added in Fig. 1d.

Figure 2

- Labeling y-axis of Figure 2A is incorrect, should just be relative expression

Thanks for pointing out. It has been changed to relative expression

Figure 3

- In the text it's stated that stimulation with AA causes rapid calcium influx, and that this peak response was smaller than TCR mediated calcium influx. There are no graphs or data to show this (especially in the phosphorylation of CAMKII).

TCR-induced phosphorylation of CAMKII was added as new Fig.3b.

- In Fig. 3B, it's unclear how the stimulation with calcium is being performed. Are stores being depleted prior or how is just adding calcium causing phosphorylation of CAMKII.

In the original figure, Ca²⁺ addition to the medium was sufficient to induce phosphorylation of CAMKII. This data was replaced by using TCR-mediated phosphorylation of CAMKII as positive control, as seen in new Fig.3b.

- In Fig. 3E, it would be useful to have a positive control for CD69 expression like CD3 stimulation to know how much CD69 is upregulated in response to TCR stimulation.

CD3/CD28 induced CD69 expression was added as positive control in new Fig. 3e.

Figure 4

- In Fig 4A, it would be critical to know if siOrai3 also reduces SOCE or if siOrai1 reduces AA induced calcium influx.

We performed silencing of Orai3 in CD4 naïve T cells and did not observe a change in SOCE-dependent calcium influx induced by Thapsigargin. See supplemental Fig. 5.

- In Figure 4B, AA stimulation is shown to induce activation of downstream signaling molecules like ERK. Do the authors think AA stimulation can induce activation of NFAT isoforms?

We expect that the AA-induced Ca²⁺ influx might induce the activation of NFAT Isoforms Our main in vitro read-out system, CD69 is relatively independent of NFAT activity. However, the inflammatory response and the cytokine production that was dampened by ORAI3 silencing is NFAT dependent, suggesting that ORAI3-induced Ca²⁺ influx function at least as a cofactor. We have included a short discussion on this subject.

- Quantifications missing for Fig. 4E and 4F.

Quantification were done in Fig.4e and 4f.

Figure 5

- Include fura trace for Fig. 5A

Fura trace was added in Fig 5A.

- It's stated that RA CD4 naïve T cells have higher basal CD69 expression compared to healthy controls, but no statistics are shown for this (only comparisons for after AA stimulation). Looking at the baseline levels for HC and RA, they don't appear much different.

We included a statistical comparison of the baseline CD69 expression between HC and RA, or HC and PsA. RA or PsA had significantly higher basal CD69 expression. See Fig 5c. and Fig 1d.

- No error bars are included for 5D. Was this a single experiment?

Three experiments are shown with error bar.

- Baseline levels of CD69 for RA samples in Fig. 5E are much higher than RA samples in Fig 5C. Reason for this discrepancy?

First, different antibodies were used to assess CD69 expression. Second, experiments in Fig. 5E and Fig. 5C were done at different time points on different FACs machine. The different voltage of the machines is one of the reason to cause this apparent discrepancy.

Figure 6

- Why were the siRNA experiments for IKAROS, MYB, and TFAP2a performed in HEK293 and not a more relevant model like Jurkat?

Orai3 is ubiquitously expressed in different human tissues. For the ease of transfection, we performed siRNA experiments in HEK293 cells as screening followed up with experiments in more relevant cells.

- Fig. 6H is not very convincing for increase in CD69 expression with IKAROS knockdown.

We performed new experiment and provide new data. In this experiment, we are using purified CD4 Naïve T cells and also rest the cells 48hrs after transfection with IKAROS siRNA. In general, the response to activation immediately after transfection is compromised, making it difficult to appreciate.

Figure 8

- One of the biggest oversights was the lack of inclusion of PBMCs transfected with siOrai3 in the NSG mouse model.

We have performed new experiments by knocking down Orai3 in CD45RO⁺ PBMCs from RA patients and adoptively transferring these cells to NSG mice implanted with human synovial tissue; the new data are

shown as new Fig.6.

- I'm not really sure what the new sequencing data really adds in Figure 8. If anything it shows that the negative regulatory role of IKAROS is relatively weak on Orai3 expression.

The RNA-seq were included in response to a previous question how widespread transcriptome changes are with reduced IKAROS. We agree with the reviewer that the changes are relatively small, at least at level of reduced expression seen in RA T cells, possibly with the exception of Orai3 and few other genes. We have moved the RNA-seq data from Fig.8 to supplemental Fig. 8.

REVIEWERS' COMMENTS

Reviewer #3 (Remarks to the Author):

None.